# Fine-Grained Cross-View Geo-Localization Using a Correlation-Aware Homography Estimator

**Xiaolong Wang**[*,1], **Runsen Xu**[3], **Zuofan Cui**[1], **Zeyu Wan**[1], **Yu Zhang**[†,1,2]

[1] College of Control Science and Engineering, Zhejiang University
[2] Key Laboratory of Collaborative sensing and autonomous unmanned systems of Zhejiang Province
[3] The Chinese University of Hong Kong

## Abstract

In this paper, we introduce a novel approach to fine-grained cross-view geo-localization. Our method aligns a warped ground image with a corresponding GPS-tagged satellite image covering the same area using homography estimation. We first employ a differentiable spherical transform, adhering to geometric principles, to accurately align the perspective of the ground image with the satellite map. This transformation effectively places ground and aerial images in the same view and on the same plane, reducing the task to an image alignment problem. To address challenges such as occlusion, small overlapping range, and seasonal variations, we propose a robust correlation-aware homography estimator to align similar parts of the transformed ground image with the satellite image. Our method achieves sub-pixel resolution and meter-level GPS accuracy by mapping the center point of the transformed ground image to the satellite image using a homography matrix and determining the orientation of the ground camera using a point above the central axis. Operating at a speed of 30 FPS, our method outperforms state-of-the-art techniques, reducing the mean metric localization error by 21.3% and 32.4% in same-area and cross-area generalization tasks on the VIGOR benchmark, respectively, and by 34.4% on the KITTI benchmark in same-area evaluation.

## 1 Introduction

Accurate localization of ground cameras is essential for various applications such as autonomous driving, robot navigation, and geospatial data analysis. In crowded urban areas, cross-view localization using satellite images has proven to have great potential for correcting noisy GPS signals [3, 38] and improving navigation accuracy [42, 13, 19]. In this paper, we consider the task of fine-grained cross-view geo-localization, which estimates the 3-Dof pose of a ground camera, *i.e.*, GPS location and orientation (yaw), from a given ground-level query image and a geo-referenced aerial image.

Previous works on coarse cross-view localization [10, 16, 25, 11, 42, 37, 41] have achieved high recall rates by formulating it as an image retrieval problem. However, the localization accuracy obtained using this method is limited by the segmentation size of satellite image patches, resulting in localization errors of tens of meters.

Recently, there has been a growing interest in fine-grained cross-view geo-localization, which assumes the availability of the ground image and a corresponding GPS-labeled satellite image patch covering the same area. Existing methods can be divided into two categories: those based on repeated sampling [27, 24, 7, 26] and those employing descriptor candidates splitting from satellite image features

---

[*] Contact: xlking@zju.edu.cn. The Code is available at `https://github.com/xlwangDev/HC-Net`.
[†] Corresponding author.

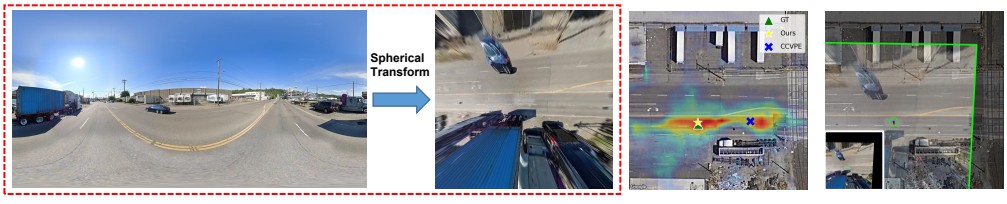

| (a) Ground Image | (b) BEV Image | (c) Satellite Image | (d) Homography Maped |

Figure 1: Visualization of the cross-view localization process in our method. The satellite image (c) is accompanied by a correlation map that represents the dense probability distribution for localization. Aligning the BEV with the satellite image using the homography matrix is shown in (d).

[36, 12, 35]. Sampling-based methods transform satellite or ground images (or feature maps) to compare with another view, involving the multiple applications of geometric transformations based on potential ground camera poses. However, these methods suffer from low localization accuracy [27, 24, 26] and prolonged time consumption[27, 24, 7, 26]. [36, 12, 35] split the features of satellite images into numerous candidates to obtain more effective and refined descriptors for comparison with the ground image descriptor. Their limited pixel-level localization resolution and high memory usage present challenges for deployment in real-world end-terminals. Hence, the pursuit of a methodology that encompasses swift computation, low memory utilization, and elevated levels of both localization accuracy and resolution remains a paramount area of research.

We observe that projecting ground images onto a bird's-eye view perspective can make satellite-based localization tasks more intuitive, similar to the way humans use maps for navigation, eliminating the need for multiple sampling or further splitting of satellite patches. In this study, we develop a spherical transform module that leverages the imaging model of ground cameras to project panoramic images onto an aerial perspective, effectively bridging the gap between distinct viewpoints. As depicted in Figure 1 (b), our approach does not necessitate prior pose information, and it facilitates the straightforward determination of the ground image's position within the satellite image (c). By transforming the complex cross-view localization problem into a 2D image alignment problem, our method enables the acquisition of precise ground GPS location and orientation.

The core problem in 2D image alignment lies in obtaining the homography transformation between two images. Given that feature-based methods like [22, 28] may not yield effective results in obstructed or unclear scenes, we employ a correlation-aware homography estimation approach. We utilize a recurrent convolutional neural network block to maximize the correlation between similar regions in the feature maps, unlike previous iterative optimization methods [24], which minimize overall feature differences. Our method ensures low correlation in unobservable areas, minimizing their impact on homography estimation, resulting in an end-to-end network that aligns the most similar parts of the transformed ground image with the satellite image, directly outputting its GPS location. Moreover, our approach has successfully overcome the challenge of lacking compact supervision information in homography estimation, *i.e.*, the absence of at least four matching point pairs. While utilizing the VIGOR dataset [42], we also address the inherent errors in the original ground truth labels to facilitate further research.

The main contributions of this paper include:

- A novel and effective method for fine-grained cross-view geo-localization, which strictly aligns ground and aerial image domains using a geometry-constrained spherical transform, reducing the problem to 2D image alignment.

- A correlation-aware homography estimation module that eliminates repeated sampling and disregards unobservable content in the images, resulting in excellent generalization performance and rapid inference speed at 30 FPS.

- Extensive experiments demonstrating that our method outperforms the state-of-the-art on two fine-grained cross-view localization benchmarks. Specifically, on the same-area and cross-area splits of the VIGOR benchmark [42], our method reduces the mean localization error by 21.3% and 32.4%, respectively. On the same-area split of the KITTI benchmark [8], our method reduces the mean localization error by 34.4%.

## 2 Related Work

**Cross-view image retrieval** methods for geo-localization, which use ground images as queries and all patches in a satellite image database as references, have been studied for years and rely on global image descriptors for successful retrieval. Early works [15, 34, 33, 31, 39] suffer from low retrieval accuracy due to significant appearance gaps and poor metric learning techniques. SAFA [25], Shi *et al*. [27], L2LTR[37], and TransGeo[41] have improved localization accuracy by employing polar transform algorithms, considering orientation, and using transformers [32].

**Fine-grained cross-view localization** beyond one-to-one retrieval is first proposed in CVR[42] and they propose a corresponding benchmark VIGOR. [27] and [24] project the satellite view using the candidate poses or the iterative optimization method and select the one that is most similar to the ground as the localization result. [7] and [26] employ transformers to acquire BEV representations of ground images, and then transform BEV maps by all possible ground camera poses to compare with the feature representations of satellite images. However, there remains untapped potential for refinement, and their inference entails prolonged time. [36] splits the features of satellite patches into several sub-volumes. [12] considers orientation and geometric information to create effective satellite descriptors for a set of candidate poses. [35] incorporates orientation and employs a coarse-to-fine manner to enhance accuracy. Despite these improvements, all three methods face resolution and high memory usage challenges stemming from the splitting or candidate selection process.

**Bird's-Eye View Perspective** has been thought of as the preferred representation for many tasks (*e.g.* navigation or localization). Recent methods [14, 7, 26] utilize transformers to project feature maps of ground images onto BEV. However, there has been no utilization of **explicit BEV images** for fine-grained cross-view geo-localization. A challenge could stem from the requirement of known camera calibration for traditional Inverse Projective Mapping methods, as [23] used. In an attempt to enhance BEV representations, Boosting [26] also explores geometric-based methodologies, but due to the necessity for depth information, transformers remain a crucial tool to handle ambiguity.

**Homography estimation** is the process of estimating a mapping between two images of a planar surface from different perspectives. DeTone *et al*. [6] is the first to propose a deep learning-based approach to homography estimation. Nie *et al*. [20] proposed the use of a global correlation layer to address the problem of small overlapping regions between the two images. Zhao *et al*. [40] incorporate the Lucas-Kanade (LK) algorithm [18] as a non-trainable iterator and combine it with CNNs. They also investigate GPS-denied navigation using Google static maps and satellite datasets based on this method. Inspired by RAFT [30], IHN [4] designed a completely iterative trainable deep homography estimation network, achieving high accuracy but requiring strong supervision. HomoGAN [9] designed an unsupervised homography estimation network based on transformer and GAN methods, achieving promising results while consuming more computational resources.

## 3 Method

This paper presents a novel approach to address the task of fine-grained cross-view localization, as illustrated in Figure 2. Our **H**omography estimation-based **C**ross-view geo-localization **Net**work, named HC-Net, takes corresponding spherical-transformed ground image and GPS-tagged satellite image as input and outputs the homography matrix between them, as well as the precise GPS location and orientation of the ground camera. The following sections introduce the spherical transform for projecting panoramas to a bird's-eye view (Section 3.1), the homography estimator for precise alignment of ground BEV images with satellite images (Section 3.2), and the supervision method of our network (Section 3.3).

### 3.1 Spherical Transform

**Panoramic imaging model** Panoramas capture a full 360-degree view on a spherical projection plane and use equirectangular projections for display, as shown in Figure 2 (a). We represent points in the ground camera coordinate system as $P = (x_1, y_1, z_1)$ and projected points in the normalized equirectangular coordinate system of the panorama as $P' = (x_2, y_2)$. The spherical coordinates $(\phi, \theta)$ are acquired from $(x_1, y_1, z_1)$ using inverse trigonometric functions, with north latitude and east longitude being positive. And the projection between the spherical coordinates $(\phi, \theta)$ and the normalised equirectangular coordinates $(x_2, y_2)$ is expressed in Figure 2 (a), (b).

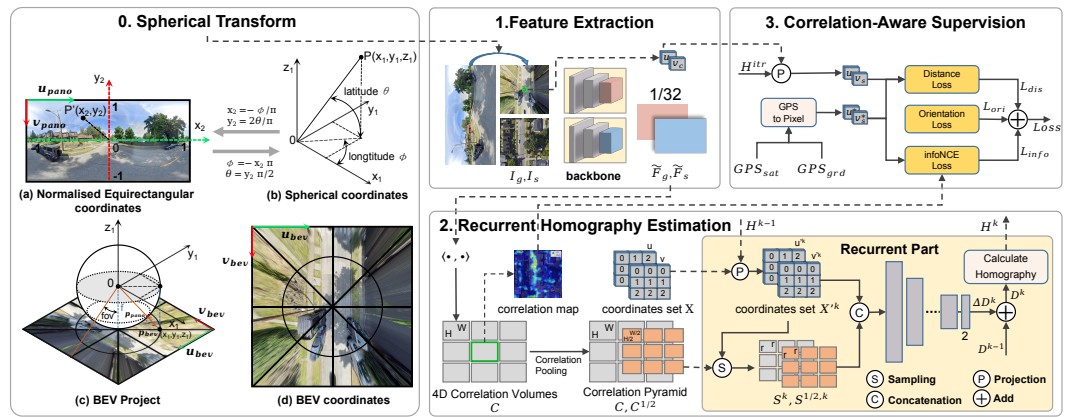

Figure 2: An overview of the proposed method for fine-grained cross-view localization. (0) illustrates the geometric principles behind our spherical transformation, while (1), (2), and (3) demonstrate our network architecture and supervision methods.

**Spherical transform**   To obtain the corresponding bird's-eye view of the panorama, we place a tangent plane at the south pole of the spherical imaging plane as a new imaging plane, as shown in Figure 2 (c). Formally, let $H_p \times W_p$ be the size of the panorama and $H_b \times W_b$ be the target size of the bird's-eye view after the spherical transform. We connect the camera's optical center with each pixel point on the BEV imaging plane, determining the corresponding pixel position in the panorama through the intersection of the connection line with the spherical imaging plane. We demonstrate the pixel coordinates on the bird's-eye view imaging plane as $(u_b, v_b)$. The camera coordinates $(x_1, y_1, z_1)$ corresponding to the pixel coordinates $(u_b, v_b)$ can be directly obtained. We set the parameter $fov = 85°$ to determine the field of view of the bird's-eye view. The focal length of the BEV in the imaging process is $f = 0.5W_b/\tan(fov)$. Therefore, each corresponding panoramic image coordinates $(u_p, v_p)$ is established as:

$$\begin{cases} u_p = [1 - \arctan2(W_b/2 - u_b, H_b/2 - v_b)/\pi] \, W_p/2, \\ v_p = [0.5 - \arctan2(-f, \sqrt{(W_b/2 - u_b)^2 + (H_b/2 - v_b)^2})/\pi]H_p. \end{cases} \quad (1)$$

After the spherical transform, the ground image is projected into the aerial image's perspective, as seen in Figure 2(d). This approach enables high-resolution localization without multiple sampling or splitting of satellite patches. Unlike the satellite-based image projection discussed in [27, 24], this type of projection does not require the selection of a projection point. We adopt a similar approach for the KITTI dataset [8] by setting up an overlooking imaging plane. This allows us to project front-view images into a bird's-eye view without any known camera calibration, as shown in Figure 4 (a), see details in Supplementary Material. Despite being applied as a preprocessing step in our method, note that the transform is differentiable. This property opens up possibilities for future research on the 5-DoF pose of the ground camera (details in Supplementary Material).

### 3.2   Homography Estimator

To achieve precise localization by aligning the transformed ground image $I_g$ with the satellite image $I_s$, we propose a correlation-based deep homography estimator inspired by IHN [4]. We resize the two images $I_g$ and $I_s$ to the same size and feed them into a Pseudo-Siamese CNN to extract features. These features are denoted as $\widetilde{F}_g \in \mathbb{R}^{D \times H \times W}$ and $\widetilde{F}_s \in \mathbb{R}^{D \times H \times W}$. As shown in Figure 2, after feature extraction, correlation computation, and recurrent homography estimation, the homography matrix $\mathbf{H}$ between $I_g$ and $I_s$ can be obtained.

**Correlation Computation**   Given the feature maps $\widetilde{F}_g$ and $\widetilde{F}_s$, the correlation volume $\mathbf{C} \in \mathbb{R}^{H \times W \times H \times W}$ is formed by taking the dot product between all pairs of feature vectors as:

$$\mathbf{C}(\widetilde{F}_g, \widetilde{F}_s) \in \mathbb{R}^{H \times W \times H \times W}, \quad C_{ijkl} = \mathrm{ReLU}(\widetilde{F}_g(i,j)^{\mathrm{T}} \widetilde{F}_s(k,l)). \quad (2)$$

To enlarge the perception range within a feature scale, we conduct average pooling on $\mathbf{C}$ at the last 2 dimensions with stride 2 to form another correlation volume $\mathbf{C}^{\frac{1}{2}} \in \mathbb{R}^{H \times W \times H/2 \times W/2}$.

**Recurrent Homography Estimation**   We refine the estimation of the homography through loop iterations. The coordinates set on $\widetilde{F}_g$ are denoted as $\mathbf{X} \in \mathbb{R}^{2 \times H \times W}$, and the coordinates set on $\widetilde{F}_s$, which is projected from $\mathbf{X}$ using the homography matrix $\mathbf{H}$, is denoted as $\mathbf{X}' \in \mathbb{R}^{2 \times H \times W}$. For each coordinate position, we denote $x = (u, v)$ in $\mathbf{X}$ and $x' = (u', v')$ in $\mathbf{X}'$. In each iteration, we use the Equation 3 to calculate $\mathbf{X}'^k$ and use it to sample the last two dimensions of $\mathbf{C}, \mathbf{C}^{\frac{1}{2}}$ with a local square grid of fixed search radius $r$, resulting in correlation slices $\mathbf{S}^k$ and $\mathbf{S}^{\frac{1}{2},k}$ of size $H \times W \times r \times r$.

$$\begin{bmatrix} u'^k \\ v'^k \\ 1 \end{bmatrix} \sim \begin{bmatrix} \mathbf{H}^k_{11} & \mathbf{H}^k_{12} & \mathbf{H}^k_{13} \\ \mathbf{H}^k_{21} & \mathbf{H}^k_{22} & \mathbf{H}^k_{23} \\ \mathbf{H}^k_{31} & \mathbf{H}^k_{32} & 1 \end{bmatrix} \begin{bmatrix} u \\ v \\ 1 \end{bmatrix} \tag{3}$$

Then we utilize a convolutional neural network module for residual homography estimation. As in [4], we parameterize the homography matrix using the displacement vectors of the 4 corner points of an image, namely the displacement cube $\mathbf{D}$. In iteration $k$, we feed the concatenated correlation slice $\mathbf{S}^k, \mathbf{S}^{\frac{1}{2},k}$, the coordinates set $\mathbf{X}$, and the currently projected coordinates set $\mathbf{X}'^k$ into the CNN module. The module is mainly composed of multiple convolutional units until the spatial resolution of the feature map reaches $2 \times 2$, where each unit downsamples the input by a scale of 2. Then a $1 \times 1$ convolutional unit projects the feature map into a $2 \times 2 \times 2$ cube $\Delta \mathbf{D}^k$, which is the estimated residual displacement vector of the 4 corner points. In iteration k, we update $\mathbf{D}^k$ by adding the estimated residual displacement vector $\Delta \mathbf{D}^k$ to $\mathbf{D}^{k-1}$. Using $\mathbf{D}^k$, we can obtain the homography matrix $\mathbf{H}^k$ through the direct linear transform [2]. The updated $\mathbf{H}^k$ is then used to project $\mathbf{X}$ in the next iteration.

## 3.3   Network Supervison

**Label Correction**   We use the VIGOR dataset [42] for fine-grained cross-view localization training and evaluation. However, the original labels in [42] contain errors up to 3 meters due to the use of approximate and consistent meter-to-pixel resolutions of the aerial images. Although [12] attempts to correct the labels, they require city-based computations or selections for resolution, which brings significant inconvenience. In our approach, we propose using the **Web Mercator projection** [1] used by virtually all major online map providers to accurately convert GPS coordinates to pixel coordinates on satellite patches, thereby enhancing generality and convenience. The main equation is as:

$$\begin{cases} x = \left[ \dfrac{256}{2\pi} 2^{zoom} (lon + \pi) \right] \text{ pixels}, \\ y = \left[ \dfrac{256}{2\pi} 2^{zoom} \left( \pi - \ln \left[ \tan \left( \dfrac{\pi}{4} + \dfrac{lat}{2} \right) \right] \right) \right] \text{ pixels}, \end{cases} \tag{4}$$

where $zoom$ indicates the zoom level for satellite images ($zoom = 20$ in VIGOR and $zoom = 19$ in KITTI) and 256 is the resolution of each satellite tile. We have applied the same method to create training labels for KITTI dataset [8], see Supplementary Material for details.

**Loss Function**   Using the homography matrix $\mathbf{H}$ obtained in Section 3.2, we project the center point of BEV onto the satellite image to obtain the localization pixel $(u_s, v_s)$. Using Equation 4, we calculated the pixel coordinates of the ground truth GPS corresponding to the satellite image as $(u_s^*, v_s^*)$. Furthermore, we use Equation 4 to project another point on the BEV centerline onto the satellite image. Connecting this point with $(u_s, v_s)$ allows us to determine the orientation of the ground camera as $\theta$.

We use a hybrid loss function, defined as $\mathcal{L} = \alpha_1 \mathcal{L}_{dis} + \alpha_2 \mathcal{L}_{ori} + \alpha_3 \mathcal{L}_{info}$, to guide our training. Here, $\mathcal{L}_{dis} = (u_s - u_s^*)^2 + (v_s - v_s^*)^2$ represents the L2 norm loss between the predicted result and the ground truth. $\mathcal{L}_{ori} = |\theta - \theta^*|$ represents the L1 norm loss between the predicted $\theta$ and the ground truth $\theta^*$. To generate a probability distribution that can be further utilized for localization, we propose the use of an infoNCE loss [21] to reinforce the correlation between the BEV point used for localization and the satellite image. $\mathcal{L}_{info}$ is defined as:

$$\mathcal{L}_{info} = -\log \frac{\exp \left( C \left( i_c, j_c, k^+, l^+ \right) / \tau \right)}{\sum_{k,l} \exp \left( C \left( i_c, j_c, k, l \right) / \tau \right)}, \tag{5}$$

where $(i_c, j_c)$ represents the center coordinates of $\widetilde{F}_g$, and $(k^+, l^+)$ represents the downsampled position of $(u_s^*, v_s^*)$ in the satellite feature map $\widetilde{F}_s$. The hyper-parameter $\tau$ is introduced to adjust the sharpness of the resulting probability distribution.

Finally, through the inverse process of Equation 4, we can determine the GPS coordinates corresponding to $(u_s, v_s)$, thereby obtaining **highly accurate localization** outputs.

# 4 Experiments

In this section, we first introduce two used datasets, evaluation metrics, and implement details of our network. We then compare the performance of our HC-Net to state-of-the-art and examine its ability to generalize to new measurements within the same areas, across different areas, and across datasets. Finally, we present ablation studies and computational efficiency analysis.

## 4.1 Datasets and Evaluation Metrics

**VIGOR dataset** [42] contains geo-tagged ground-level panoramas and aerial images collected in four cities in the US. Each aerial patch corresponds to a ground area of approximately $70m \times 70m$. From Figure 1, it can be observed that the effective field of view of the ground image is slightly smaller than this range. A patch is considered positive if its center $1/4$ region contains the ground camera's location, otherwise, it is semi-positive. In our experiments, we use positive aerial images for training and testing all models. The panoramas are shifted based on orientation information to align North in the middle, indicating that the orientation prior is known. During training, we introduce $\pm45°$ noise to the orientation prior to generating orientation labels. The dataset provides $105, 214$ panoramas for the geo-localization experiments. We adopt the Same-Area and Cross-Area splits from [42]. For validation and hyperparameter tuning, we randomly select 20% of the data from the training set, as done in[36, 12, 35]. Compared to [12], we have more accurately corrected the ground truth labels in [42], as mentioned in Section 3.3.

**KITTI dataset** [8] contains ground-level images captured by a moving vehicle with a forward-facing viewpoint, which is a restricted viewpoint. [24] augments the dataset with aerial images. Each aerial patch corresponds to a ground area of approximately $100m \times 100m$. The Training and Test1 sets consist of different measurements from the same region, while the Test2 set has been captured in a different region. As assumed in [24], ground images are located within a $40 \times 40m$ area in the center of the corresponding aerial patches, and there is an orientation prior with noise between $\pm10°$.

**Evaluation metrics** in our experiments are the mean and median error between the predicted and ground truth over all samples for both localization and orientation separately in meters and in degrees. Following [35], for the KITTI dataset, we additionally include the recall under a certain threshold for longitudinal (driving direction) and lateral localization error, and orientation estimation error. Our thresholds are set to 1m and 5m for localization and to $1°$ and $5°$ for orientation estimation.

## 4.2 Implementation Details

Our network uses EfficientNet-B0 [29] with pretrained weights on Imagenet [5] as both the ground and aerial feature extractors, with non-shared weights. The satellite image and bird's-eye-view (BEV) transformed from the ground image both have a size of $512 \times 512$ on both the VIGOR [42] and KITTI [8] datasets. PyTorch is used for network implementation, and training is done using the AdamW [17] optimizer with a maximum learning rate of $3.5 \times 10^{-4}$. The network is trained with a batch size of 16 and a training iteration of 180000. We set the search radius of the correlation updater $r = 4$ and set $\alpha_1 = 0.1, \alpha_2 = 10, \alpha_3 = 1.0, \tau = 4$ in the loss function. The total iteration $K = 6/10$ and the feature map size $D \times H \times W = 320 \times 16 \times 16/320 \times 16 \times 16$ on VIGOR/KITTI.

## 4.3 Comparison with State-of-the-Art Methods

**VIGOR dataset**   On the **VIGOR** dataset, we compare our method against several state-of-the-art methods: CVR [42], SliceMatch [12], Boosting [26] and CCVPE [35]. For the performance evaluations of CVR [42], SliceMatch [12], and Boosting [26] with known orientation priors, we directly utilize the results provided by CCVPE or the corresponding paper. Regarding most state-of-the-art CCVPE [35], we executed its official code to obtain results not presented in the original

Table 1: Location and orientation estimation error on VIGOR [42] dataset. **Best in bold.** Different levels of noise are added to the orientation prior. "*" indicates methods using our corrected labels.

| Noise | Method | Same-Area ↓Localization(m) mean | median | ↓Orientation(°) mean | median | Cross-Area ↓Localization(m) mean | median | ↓Orientation(°) mean | median |
|---|---|---|---|---|---|---|---|---|---|
| 0° | CVR [42] | 8.82 | 7.68 | - | - | 9.45 | 8.33 | - | - |
| | SliceMatch [12] | 5.18 | 2.58 | - | - | 5.53 | 2.55 | - | - |
| | Boosting [26] | 4.12 | 1.34 | - | - | 5.16 | **1.40** | - | - |
| | CCVPE [35] | 3.60 | 1.36 | 10.59 | 5.43 | 4.97 | 1.68 | 27.78 | 14.11 |
| | CCVPE* [35] | 3.37 | 1.33 | 9.47 | 5.23 | 4.96 | 1.69 | 26.30 | 14.21 |
| | HC-Net(ours)* | **2.65** | **1.17** | **1.92** | **1.04** | **3.35** | 1.59 | **2.58** | **1.35** |
| ±20° | CCVPE* [35] | 3.48 | 1.39 | 9.80 | 5.49 | 5.16 | 1.78 | 26.36 | 14.87 |
| | HC-Net(ours)* | **2.65** | **1.17** | **1.93** | **1.02** | **3.36** | **1.59** | **2.62** | **1.34** |
| ±45° | CCVPE* [35] | 3.50 | 1.39 | 10.56 | 5.95 | 5.16 | 1.78 | 26.77 | 15.29 |
| | HC-Net(ours)* | **2.70** | **1.18** | **2.12** | **1.04** | **3.46** | **1.60** | **3.00** | **1.35** |

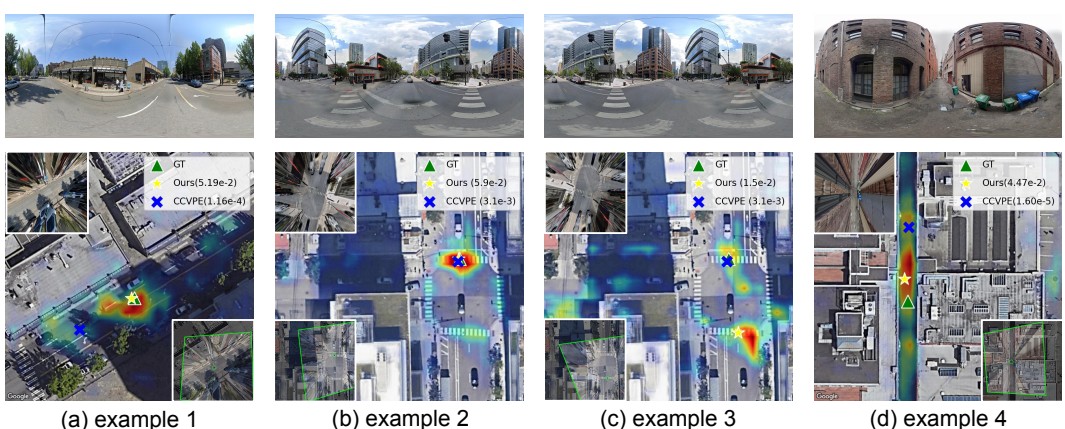

(a) example 1   (b) example 2   (c) example 3   (d) example 4

Figure 3: Visualization of our method on the VIGOR [42] dataset with localization confidence probability indicated in the legend. (d) demonstrates a failure case of our method when dealing with scenes that are radially monotonous and repetitive-uninformative.

paper and used the published results where applicable. We also re-train and evaluate CCVPE with our corrected labels to ensure a fairer comparison. We conduct a comprehensive evaluation with different levels of noise in the orientation prior to ground images, including 0°, ±20°, ±45°.

As shown in Table 1, our method outperforms all previous methods in both same-area and cross-area settings in terms of the mean localization error metric, which suggests that our method can cope with more challenging scenarios, as shown in Figure 3(a). The mean localization error is reduced by 21.3% and 32.4% respectively. The previously best-performing method [35] shows a notable performance gap between same-area and cross-area settings, while our method significantly narrows this gap. This suggests that our method possesses superior generalization capabilities. We further demonstrate our superior generalization ability across datasets in Section 4.4.

When prior information exhibits varying degrees of noise, our method still outperforms CCVPE [35] and provides a more accurate estimation of the orientation. Moreover, our network generates confidence probabilities for localization results, allowing us to filter out potential misestimations. As shown in Figure 3 (c), when the scene contains a symmetric layout like zebra crossings and noise exceeds 90°, the homography estimation may incorrectly align the ground image with the satellite map. However, by rotating the BEV image four times and selecting the prediction with the highest confidence probability, we can obtain the correct result as in Figure 3 (b). Despite the mislocalized point having a similar appearance to the ground observation, its probability output is significantly smaller than the localization confidence probability from the correct location. This property is crucial for safety-critical applications like autonomous driving.

Table 2: Location and orientation estimation error on KITTI [8] dataset. **Best in bold.** Long. and Orien. are abbreviations for Longitudinal and Orientation, respectively.

| Method | Area | ↓Location(m) Mean | Median | ↑Lateral(%) R@1m | R@5m | ↑Long.(%) R@1m | R@5m | ↓Orien.(°) Mean | Median | ↑Orien.(%) R@1° | R@5° |
|---|---|---|---|---|---|---|---|---|---|---|---|
| LM [24] | Same | 12.08 | 11.42 | 35.54 | 80.36 | 5.22 | 26.13 | 3.72 | 2.83 | 19.64 | 71.72 |
| SliceMatch [12] | Same | 7.96 | 4.39 | 49.09 | 98.52 | 15.19 | 57.35 | 4.12 | 3.65 | 13.41 | 64.17 |
| Boosting [26] | Same | - | - | 76.44 | 98.89 | 23.54 | 62.18 | - | - | **99.10** | **100.00** |
| CCVPE [35] | Same | 1.22 | 0.62 | 97.35 | 99.71 | 77.13 | 97.16 | 0.67 | 0.54 | 77.39 | 99.95 |
| HC-Net(Ours) | Same | **0.80** | **0.50** | **99.01** | **99.73** | **92.20** | **99.25** | **0.45** | **0.33** | 91.35 | 99.84 |
| LM [24] | Cross | 12.58 | 12.11 | 27.82 | 72.89 | 5.75 | 26.48 | 3.95 | 3.03 | 18.42 | 71.00 |
| SliceMatch [12] | Cross | 13.50 | 9.77 | 32.43 | 76.44 | 8.30 | 35.57 | 4.20 | 6.61 | 46.82 | 46.82 |
| Boosting [26] | Cross | - | - | 57.72 | 91.16 | 14.15 | 45.00 | - | - | **98.98** | **100.00** |
| CCVPE [35] | Cross | 9.16 | **3.33** | 44.06 | 90.23 | 23.08 | 64.31 | **1.55** | **0.84** | 57.72 | 96.19 |
| HC-Net(Ours) | Cross | **8.47** | 4.57 | **75.00** | **97.76** | **58.93** | **76.46** | 3.22 | 1.63 | 33.58 | 83.78 |

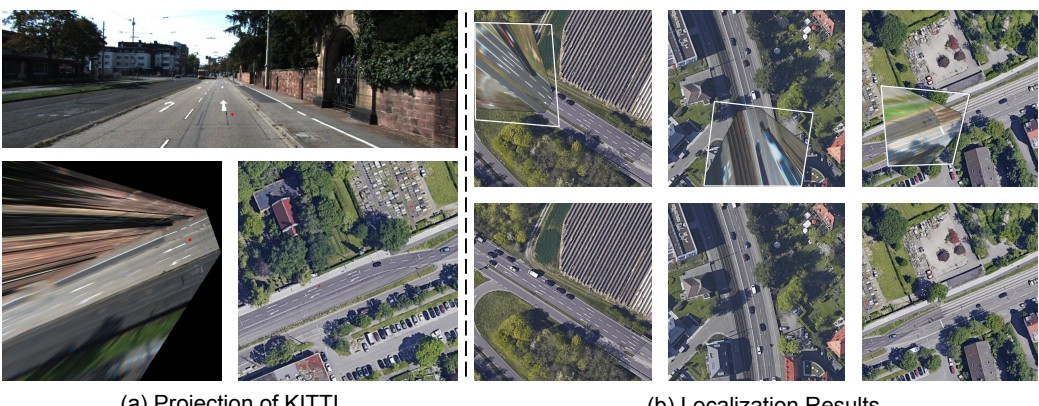

(a) Projection of KITTI        (b) Localization Results

Figure 4: (a) depicts the projection of the frontal perspective into a bird's-eye view in the KITTI dataset [8], accompanied by the corresponding satellite imagery. (b) visualizes our method's localization results on KITTI [8]: the top row shows aligned ground and satellite images using our estimated homography matrix; the bottom row presents original satellite images for reference.

**KITTI dataset** On the **KITTI** dataset, we compare our method with the state-of-the-art LM [24], SliceMatch [12], Boosting [26] and CCVPE[35]. Similar to the evaluation on VIGOR, we directly use the results provided by CCVPE [35] or the corresponding paper. We utilize an increased number of iteration steps in response to the limited coverage of ground images from KITTI [8] within satellite images. The results are shown in Table 2. When a $\pm 10°$ orientation prior is considered in both Training and Test1, our method has a 34.4% lower mean error for localization than CCVPE [35]. Figure 4 presents the visualization results on the KITTI dataset. Boosting [26] achieves exceptional accuracy in estimating orientation. This may be attributed to their utilization of a two-stage approach, where the first stage is dedicated to orientation estimation.

## 4.4 Cross-Dataset Generalization

The CVUSA dataset [39] is commonly used for cross-view localization, but it only provides retrieval labels and cannot be used for training precise localization models. Despite the panoramas in CVUSA being cropped and having a non-standard aspect ratio, our spherical transform method successfully projects them to a bird's-eye view by completing it to the correct proportion, as shown in Figure 5. We use images from CVUSA in different cities to test our model's generalization ability, which is trained on VIGOR. The alignment results in Figure 5 demonstrate that our model has strong potential and can align significantly different BEV images with their corresponding satellite images for precise localization, even in unfamiliar cities without retraining.

Table 3: Ablation Study on Pseudo-Siamese Backbone and Homography Estimation Module.

| Experiment | Mean Error | Median Error |
|---|---|---|
| Non-shared weights | **2.65 m** | **1.17 m** |
| Shared weights | 3.36 m | 1.36 m |
| SuperGlue [22] | 13.06 m | 3.76 m |
| LoFTR [28] | 28.85 m | 4.89 m |

Table 4: Comparison of computational efficiency between our method and CCVPE [35].

| | CCVPE [35] | HC-Net(ours) |
|---|---|---|
| Parameters | 57.40 M | **11.21 M** |
| Memory Usage | 4730 MiB | **1900 MiB** |
| Inference Time | 33 ms | **31 ms** |
| Mean Error | 3.37 m | **2.65 m** |

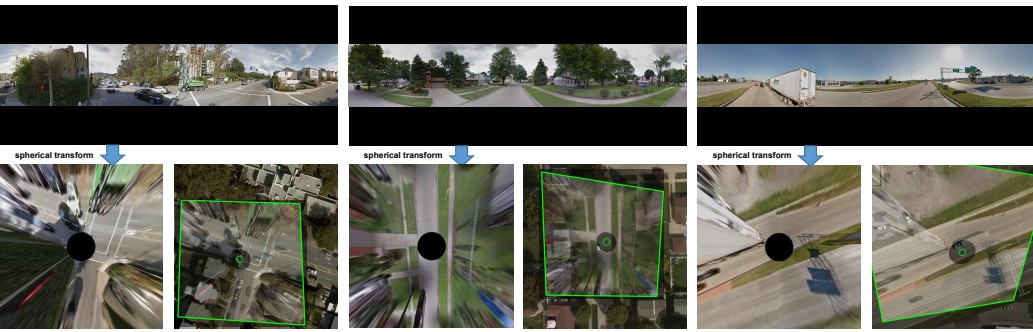

Figure 5: Three examples in CVUSA[39]. The top of each example shows the completed panorama. The bottom-left shows the bird's-eye view obtained using the spherical transform. The bottom-right shows the alignment result of our method between the BEV and the satellite image.

## 4.5   Ablation Study

**Pseudo-Siamese Backbone**   We employ a Pseudo-Siamese backbone with non-shared weights for processing images from two different sources. To demonstrate the effectiveness of this structure, we train two models with shared and non-shared weights. The results in Table 3 show that applying the Pseudo-Siamese backbone leads to a mean localization error reduction of 0.71m.

**Homography Estimation Module**   We also conduct ablation experiments to demonstrate the effectiveness of our homography estimation module. We replace the homography estimation module with feature-based methods SuperGlue [22] and LoFTR [28], utilizing RANSAC-based functions from OpenCV to compute homography matrix from matched feature points. The experimental results are presented in Table 3. The fact that the feature-based methods have low median localization errors demonstrates the effectiveness of transforming localization into 2D alignment. However, the significantly lower mean errors achieved by our proposed deep homography estimator indicate its superiority in achieving more accurate localization.

## 4.6   Computational Efficiency Analysis

We assess our method's computational efficiency against the state-of-the-art CCVPE [35]. Table 4 compares model parameters, inference memory, per-frame inference time, and mean localization error on the VIGOR dataset using a 12th Gen Intel(R) Core(TM) i5-12490F processor, 16GB memory, and an NVIDIA RTX 3050 GPU. Our method achieves higher accuracy, faster speed, and lower memory usage, demonstrating its computational efficiency.

## 5   Conclusion

In this study, we introduce HC-Net, an end-to-end network designed for fine-grained cross-view geo-localization. The network processes spherical-transformed ground images and GPS-tagged satellite images as inputs and generates the homography matrix between them, along with the precise GPS location and orientation of the ground camera. Compared to the previous state-of-the-art, HC-Net demonstrates a reduction in mean localization error by 21.3% and 32.4% on the same-area and cross-area splits of the VIGOR dataset, respectively.

## Acknowledgements

This work was supported by NSFC 62088101 Autonomous Intelligent Unmanned Systems, in part by TI 2030-Major Projects 2021ZD0201403.

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

# Supplementary Material for
# Fine-Grained Cross-View Geo-Localization Using a Correlation-Aware Homography Estimator

In this **supplementary material**, we provide detailed information on projecting ground images to bird's-eye view for VIGOR [42] and KITTI [8] datasets in Section A, as well as the computation details for converting GPS labels to pixel coordinates used during training in Section B. We also discuss the potential for future research based on our method in Section C and broader societal and industry impact of our approach in Section D.

## A  Projection Details

### A.1  Details for Spherical Transform

In the main paper, we provide an overview of the derivation process of the Spherical Transform and the final result in Equation 1. Here we will provide detailed information on the process for the projection. We will use the expressions defined in Section 3.1. To clarify the presentation, we also provide the schematic diagram of the projection process, as shown in Figure 6.

In the camera coordinates, the conversion formula between the Cartesian coordinates $P = (x_1, y_1, z_1)$ and the spherical coordinates $(\phi, \theta)$ is given by:

$$\begin{cases} \phi = \arctan2\left(y_1, x_1\right) & \in [-\pi, \pi], \\ \theta = \arctan2\left(z_1, \sqrt{x_1^2 + y_1^2}\right) & \in [-\pi/2, \pi/2]. \end{cases} \tag{6}$$

The equirectangular projection is used to project spherical coordinates onto a plane, which is the display format for panoramic images. The conversion formula between the spherical coordinates $(\phi, \theta)$ and the Normalised Equirectangular coordinates $P' = (x_2, y_2)$ is given by:

$$\begin{cases} x_2 = \dfrac{-\phi}{\pi} & \in [-1, 1], \\ y_2 = \dfrac{\theta}{\pi/2} & \in [-1, 1]. \end{cases} \tag{7}$$

The negative sign in the $x_2$ expression is due to the fact that panoramic images display the scene as viewed from the camera optical center to the outside. Therefore, when $\phi$ is positive, it corresponds to the negative half-plane of the Equirectangular plane in terms of $x_2$. The mapping between pixel coordinates $(u_p, v_p)$ and Normalised Equirectangular coordinates $(x_2, y_2)$ on the panorama is established as:

$$\begin{cases} u_p = (x_2 + 1)W_p/2 & \in [0, W_p], \\ v_p = (-y_2 + 1)H_p/2 & \in [0, H_p]. \end{cases} \tag{8}$$

To obtain the corresponding bird's-eye view of the panorama, we place a tangent plane at the south pole of the spherical imaging plane as a new imaging plane, as shown in Figure 6 (c). The focal length of the BEV in the imaging process is $f = 0.5W_b/\tan(fov)$. The camera coordinate system coordinates $(x_1, y_1, z_1)$ corresponding to a pixel $(u_b, v_b)$ on the bird's-eye view (BEV) imaging plane are determined as:

$$\begin{cases} x_1 = -v_b + H_b/2, \\ y_1 = -u_b + W_b/2, \\ z_1 = -f. \end{cases} \tag{9}$$

By substituting Equation 9 into Equation 6, and then substituting the result into Equation 7, we can obtain the Normalised Equirectangular coordinates $(x_2, y_2)$ for a pixel on the bird's-eye view (BEV) image plane. Finally, by substituting the Normalised Equirectangular coordinates into Equation 8, we can obtain the pixel coordinates $(u_p, v_p)$ on the panoramic image. This leads to the Spherical Transform mapping formula in Equation 1.

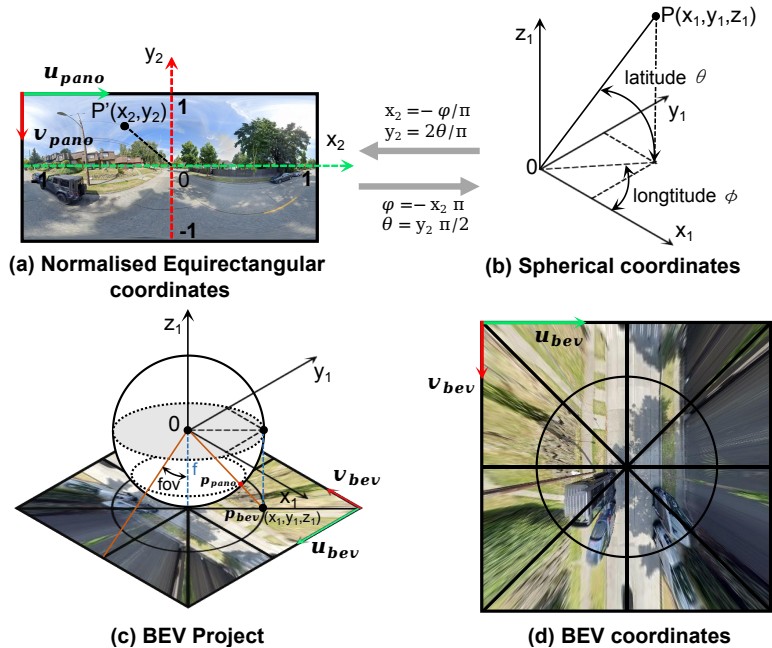

(a) Normalised Equirectangular coordinates

(b) Spherical coordinates

(c) BEV Project

(d) BEV coordinates

Figure 6: Illustration of panorama imaging model and the spherical transform mechanism for projecting panoramas to a bird's-eye view.

We test the effectiveness of the bird's-eye view projection with different field of view ($fov$) parameters, as shown in Figure 7. As seen in (d), when $fov = 85°$, the field of view of the bird's-eye view image is similar to that of the corresponding satellite image, and there are few invalid parts in the image. Therefore, we use this parameter for all of our experiments.

The spherical transform projects the ground image to a bird's eye view, while the polar transform [25] or projective transform [24] projects the satellite image to the perspective of the ground image (aligning with a panoramic view). There is a growing recognition and research on the role of Bird's Eye View (BEV) representation in localization, navigation, and related tasks.

The advantage of projecting to a bird's eye view, as compared to projecting the satellite image to the perspective of the ground image, lies in its more intuitive nature for the localization task and the need for only one projection at the start of the localization process. On the other hand, the latter approach requires selecting projection points (initial pose or candidate poses) for projecting the satellite image, and if these points are not accurately chosen, the resulting projection may not align well with the ground image. Additionally, for more accurate localization, the latter approach requires multiple applications of the projection method from different positions, which could potentially make it less real-time compared to our method.

### A.2 Details for Projecting Front View Images in KITTI

In the KITTI dataset [8], the ground is a front view image, so the process of projecting it onto a bird's-eye view is completely different from that of the panorama in VIGOR [42]. We illustrate the mechanism of this projection process in Figure 8, where $AB$ represents the height of the frontal view image, and $AC$ represents the desired height of the bird's-eye view image.

We define the size of the front view image as $H_f \times W_f$, and the target size of the bird's-eye view image as $H_b \times W_b$. Using a method similar to that described in Section 3.1, we imagine a bird's-eye projection plane parallel to the ground and adjacent to the front view image. Then we connect the camera optical center with each pixel on the bird's-eye view image to obtain the corresponding pixel coordinates on the front view image.

In Figure 8 (b), $fov$ represents the field of view of the front view image ($\angle ADC$), $f$ represents the focal length $DE$, $h$ represents the distance from the camera optical center $D$ to the bird's-eye

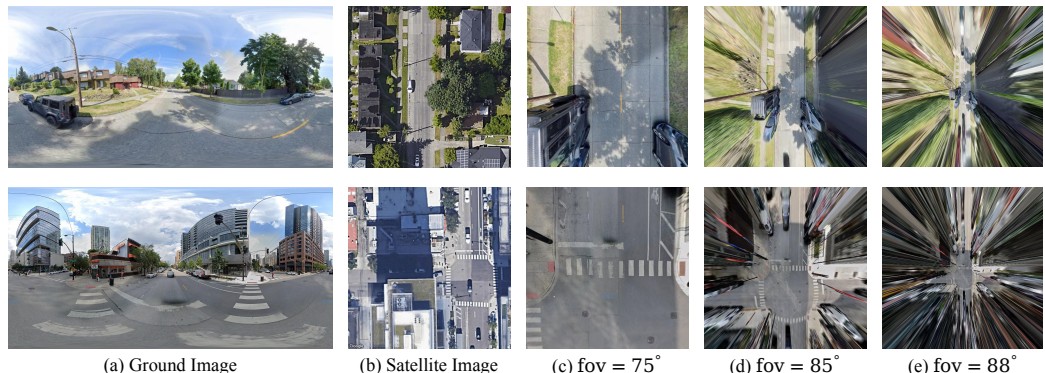

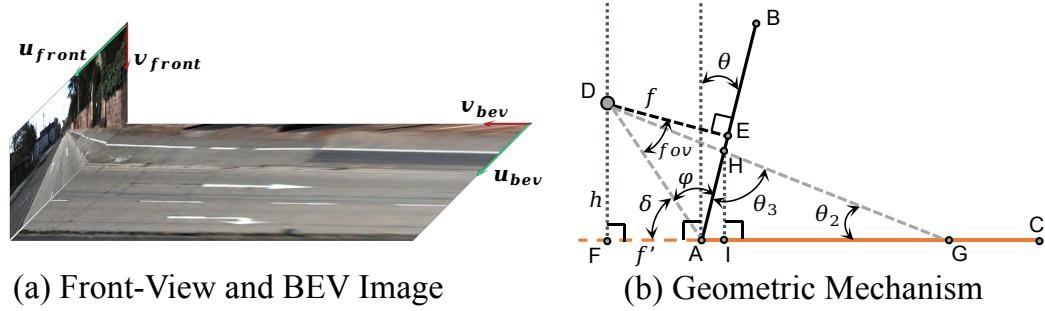

| (a) Ground Image | (b) Satellite Image | (c) fov = 75° | (d) fov = 85° | (e) fov = 88° |

Figure 7: The examples of bird's-eye view images obtained through the Spherical Transform at different field of view ($fov$) parameters.

(a) Front-View and BEV Image

(b) Geometric Mechanism

Figure 8: Illustration of the projection mechanism for KITTI [8].

projection plane, and $\theta$ represents the angle between the camera imaging plane $AB$ and the vertical direction. To facilitate the subsequent derivation, we also define $\delta$ to represent $\angle FAD$, $\varphi$ to represent $\angle BAD$, $\theta_2$ to represent $\angle AGD$, $\theta_3$ to represent $\angle AHG$, $f'$ to represent $FA$, and $l_0$ to represent $AD$. According to [8], $fov = 17.5°$ in the KITTI dataset. These variables are calculated as follows:

$$
\begin{cases}
f = \dfrac{H_f}{2} / \tan(fov) \\
\varphi = \dfrac{\pi}{2} - fov \\
\delta = \dfrac{\pi}{2} - (\varphi - \theta) \\
l_0 = \sqrt{f^2 + \left(\dfrac{H_f}{2}\right)^2} \\
h = l_0 \sin \delta \\
f' = l_0 \cos \delta.
\end{cases}
\tag{10}
$$

We denote the pixel coordinates on the bird's-eye view imaging plane as $(u_b, v_b)$. The values of $\theta_2$ and $\theta_3$ can be calculated using the arctangent formula and the properties of exterior angle as follows:

$$
\begin{cases}
\theta_2 = \arctan\left(h / \left(f' + H_b - v_b\right)\right) \\
\theta_3 = \dfrac{\pi}{2} + \theta - \theta_2.
\end{cases}
\tag{11}
$$

Then, based on the sine and similarity triangle theorems, we can calculate the corresponding pixel coordinates $(u_f, v_f)$ on the front view image for each pixel on the bird's-eye view image. The calculation is as follows:

$$
\begin{cases}
\dfrac{H_f - v_f}{H_b - v_b} = \dfrac{\sin \theta_2}{\sin \theta_3} \\
\dfrac{W_f/2 - u_f}{W_b/2 - u_b} = \dfrac{f' + (H_f - v_f) \sin \theta}{f' + H_b - v_b}
\end{cases}
\tag{12}
$$

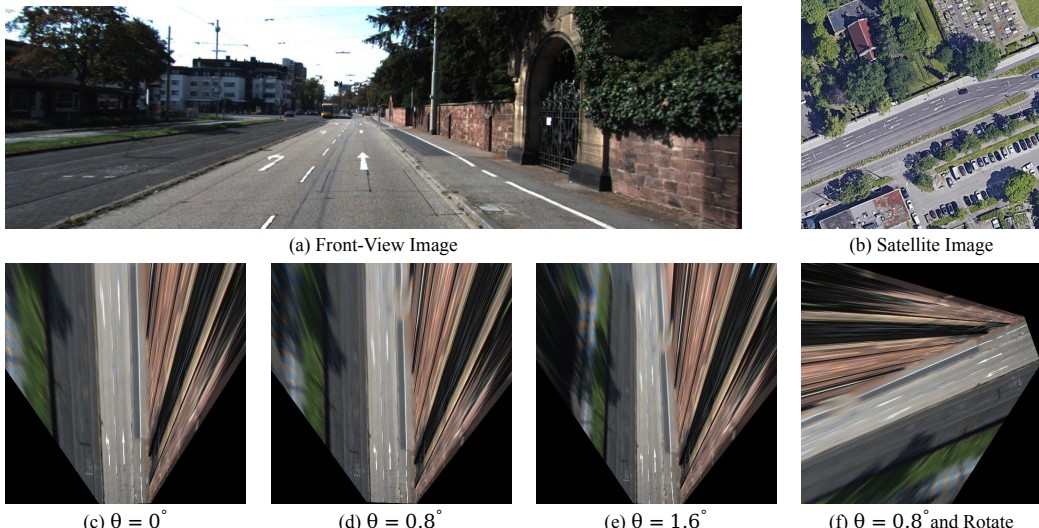

(a) Front-View Image
(b) Satellite Image

(c) $\theta = 0^{\circ}$     (d) $\theta = 0.8^{\circ}$     (e) $\theta = 1.6^{\circ}$     (f) $\theta = 0.8^{\circ}$ and Rotate

Figure 9: The examples of bird's-eye view images obtained from front view image at different $\theta$ parameters.

$$\Rightarrow \begin{cases} v_f = H_f - \dfrac{\sin\theta_2}{\sin\theta_3}(H_b - v_b) \\ u_f = \dfrac{W_f}{2} - \dfrac{f' + (H_f - v_f)\sin\theta}{f' + H_b - v_b}\left(\dfrac{W_b}{2} - u_b\right) \end{cases} \tag{13}$$

We introduce the angle $\theta$ because we found that in practical testing, the imaging plane of the ground camera may not be perfectly vertical, as shown in Figure 9. Based on experimental results, we use $\theta = 0.8°$ for all experiments. During the actual projection process, we set the target size of the bird's-eye view image to be $H_b \times W_b = (6 * W_f) \times (6 * W_f)$, which corresponds to a field of view of approximately $40m \times 40m$. After the first projection calculation, we directly applied the homography matrix $H$ obtained from the above projection process to obtain the BEV. Then, we used scaling and rotation homography matrices $H_1$ and $H_2$ to obtain the final BEV with the specified size.

$$H_1 = \begin{bmatrix} \text{scale} & 0 & 0 \\ 0 & \text{scale} & 0 \\ 0 & 0 & 1 \end{bmatrix}, H_2 = \begin{bmatrix} \cos\gamma & -\sin\gamma & u_c(1-\cos\gamma) + v_c\sin\gamma \\ \sin\gamma & \cos\gamma & v_c(1-\cos\gamma) - u_c\sin\gamma \\ 0 & 0 & 1 \end{bmatrix} \tag{14}$$

Here, scale represents the ratio of the current $H_b \times W_b$ to the desired input size of the network ($512 \times 512$). $\gamma$ represents the yaw angle of the ground camera with noise, and $(u_c, v_c)$ represent the pixel coordinates of the center of the final BEV image.

## B  Label Correction

### B.1  Label Correction in VIGOR Dataset

During research, we found that the ground truth for pixel coordinates of ground images on satellite patches in the VIGOR dataset [42] is inaccurate. The VIGOR dataset uses a uniform meter-to-pixel resolution for converting the latitude and longitude of ground images to their location in aerial images across all four cities. SliceMatch [12] also discovered this issue and proposed a correction, but their improvement is limited to using different average meter-to-pixel resolutions for each city, which may still result in some inaccuracies in different regions of the same city.

We propose the use of Mercator projection [1] to directly compute the pixel coordinates of ground images on specified satellite images using the GPS information provided in the dataset. Equation 4 allows us to calculate the pixel coordinates of a GPS coordinate on a global scale. We compute the

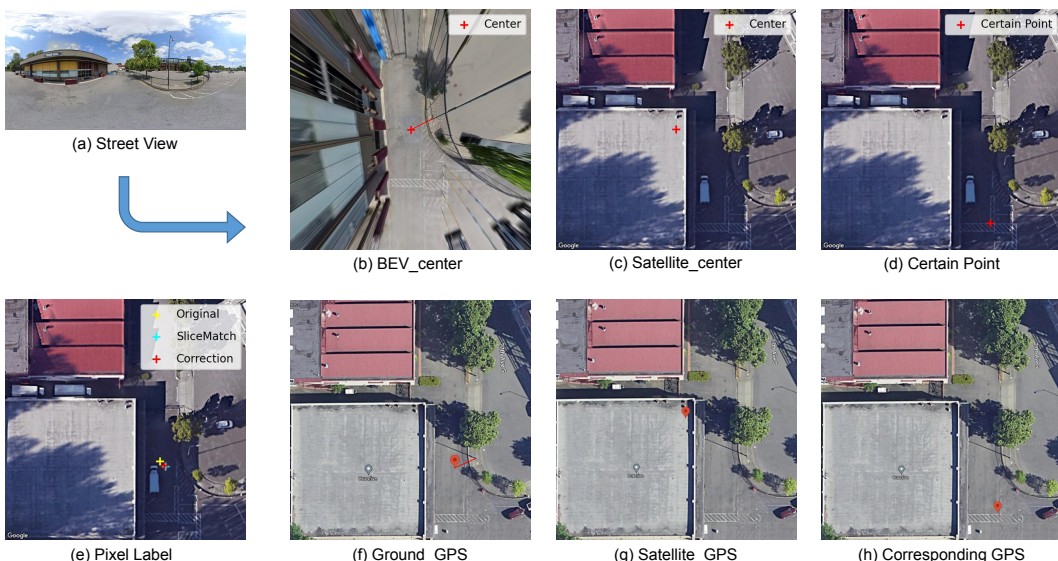

Figure 10: Illustration of label problem in VIGOR dataset [42]. In (e), we show the label provided by the VIGOR [42], the label corrected by SliceMatch [12], and the pixel coordinates computed using our method. The space distance between label provided by VIGOR and ours reaches 1.93 m. The pair of (d) and (h) represent the the pixel coordinates obtained by our method and their corresponding GPS location on Google Maps, demonstrating the accuracy of our method.

Table 5: Absolute error statistics for labels in VIGOR [42] and SliceMatch [12] in four cities. The absolute error is defined as the distance between the original and the corrected locations.

| City | VIGOR (m) | | | | SliceMatch (m) | | | |
|---|---|---|---|---|---|---|---|---|
| | Min. | Mean | Median | Max. | Min. | Mean | Median | Max. |
| Chicago | 0.00 | 0.44 | 0.45 | 0.80 | 0.00 | 0.10 | 0.13 | 0.31 |
| New York | 0.00 | 0.20 | 0.21 | 0.39 | 0.00 | 0.16 | 0.16 | 0.42 |
| San Francisco | 0.00 | 0.42 | 0.45 | 0.86 | 0.00 | 0.09 | 0.13 | 0.27 |
| Seattle | 0.00 | 1.73 | 1.80 | 3.13 | 0.00 | 0.12 | 0.13 | 0.33 |

pixel coordinates of the satellite patch's GPS label and the pixel coordinates of the ground image's GPS label. We then add the difference between them to the center coordinates of the current satellite patch, resulting in accurate pixel coordinates of the ground image on the satellite patch. The result of the correction is shown in Figure 10, which shows that the label provided by VIGOR has a significant deviation from the true position. The label corrected by SliceMatch [12] reduces this deviation, but there is still some error between the corrected label and the true position. The statistical information on the absolute error, measured in meters, between the labels provided in VIGOR [42] and SliceMatch [12] and the corrected labels using our method is presented in Table 5.

Note that in the model training process, including our method and all other methods, we do not directly use the GPS coordinates for training. This is because if GPS is used directly to calculate the training loss, the data truncation problem will occur when using Float32 tensor format since the valid data is five decimal places and trigonometric functions are used in the calculation.

## B.2 Label Creation in KITTI Dataset

Supervised information required for training our network includes the pixel location on the bird's-eye view image, as well as its corresponding true GPS value. However, obtaining this information directly from the KITTI dataset [8] is not feasible, and we need to calculate it ourselves.

To obtain the required supervised information, we use the calibration files provided by KITTI and the point cloud data in the dataset. Given a 3D point $\mathbf{x}$ in Velodyne coordinates, we project it to a point $\mathbf{y}$

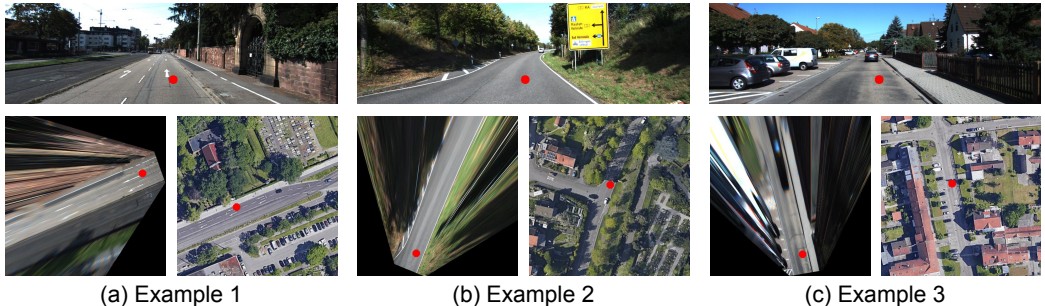

|       (a) Example 1       |       (b) Example 2       |       (c) Example 3       |

Figure 11: Three examples of labels created for the KITTI dataset [8]. The red dots indicate the location of the same point in the front view, bird's-eye view, and satellite view, respectively.

in the i-th camera image as follows:

$$\mathbf{y} = \mathbf{P}_{\text{rect}}^{(i)} \mathbf{R}_{\text{rect}}^{(0)} \mathbf{T}_{\text{velo}}^{\text{cam}} \mathbf{x}, \tag{15}$$

where $i = 2$, $\mathbf{P}\text{rect}^{(i)}$, $\mathbf{R}\text{rect}^{(0)}$, and $\mathbf{T}_{\text{velo}}^{\text{cam}}$ are calibration parameters provided in the KITTI dataset.

Additionally, we calculate the distance deviation in terms of latitude and longitude for the 3D point with respect to the origin of the IMU/GPS coordinate system in the world coordinate system. This is represented as $\mathbf{x}_{\text{GPS}}$:

$$\mathbf{x_{GPS}} = \mathbf{T}_{\text{imu}}^{\text{w}} \mathbf{T}_{\text{velo}}^{\text{imu}} \mathbf{x}. \tag{16}$$

Using the above process, we compute the latitude and longitude deviations for a particular 3D point relative to the GPS sensor, and determine its corresponding GPS value using the meter-to-GPS calculation method. Finally, we utilize the homography transformation $H_{\text{final}} = H_2 H_1 H$ obtained from Section A.2 to map the pixel coordinates $\mathbf{y}$ to their corresponding pixel coordinates on the BEV image. During model inference, we can obtain the GPS coordinates of a particular pixel on the BEV image. Using the inverse process of the above method, we can obtain the latitude and longitude deviations of the corresponding 3D point relative to the GPS sensor, and estimate the GPS value corresponding to the camera. Some examples is shown in Figure 11.

## C   Future Research

In our projection of the panoramic images into a bird's-eye view, we place a tangent plane at the south pole of the spherical imaging plane as the new imaging plane. However, this method assumes that the ground camera is oriented vertically, which may not always be the case due to roll and pitch deviations. Assuming $(\alpha, \beta, \gamma)$ represent the angles of $(roll, pitch, yaw)$, the rotation matrix can be obtained as follows:

$$\mathbf{R} = \begin{bmatrix} \cos\gamma\cos\beta & \cos\gamma\sin\beta\sin\alpha - \sin\gamma\cos\alpha & \cos\gamma\sin\beta\cos\alpha + \sin\gamma\sin\alpha \\ \sin\gamma\cos\beta & \sin\gamma\sin\beta\sin\alpha + \cos\gamma\cos\alpha & \sin\gamma\sin\beta\cos\alpha - \cos\gamma\sin\alpha \\ -\sin\beta & \cos\beta\sin\alpha & \cos\beta\cos\alpha \end{bmatrix}. \tag{17}$$

By multiplying the rotation matrix $\mathbf{R}$ with the camera coordinates obtained from Equation 9, we obtain the new coordinates $(x_1', y_1', z_1')$ in the camera coordinate system. We then proceed with the subsequent calculations to obtain the bird's-eye view image under the assumption of the specified $(roll, pitch, yaw)$ angles. Different angle configurations yield different bird's-eye view images, as illustrated in Figure 12. It can be observed that the resulting BEV appearance varies significantly when different $(roll, pitch)$ angles are given. Therefore, we suggest that in future research, $(roll, pitch)$ can be treated as additional predicted outputs to obtain more degrees of freedom in estimating the ground camera pose. Our differentiable spherical transform provides a possibility for this idea.

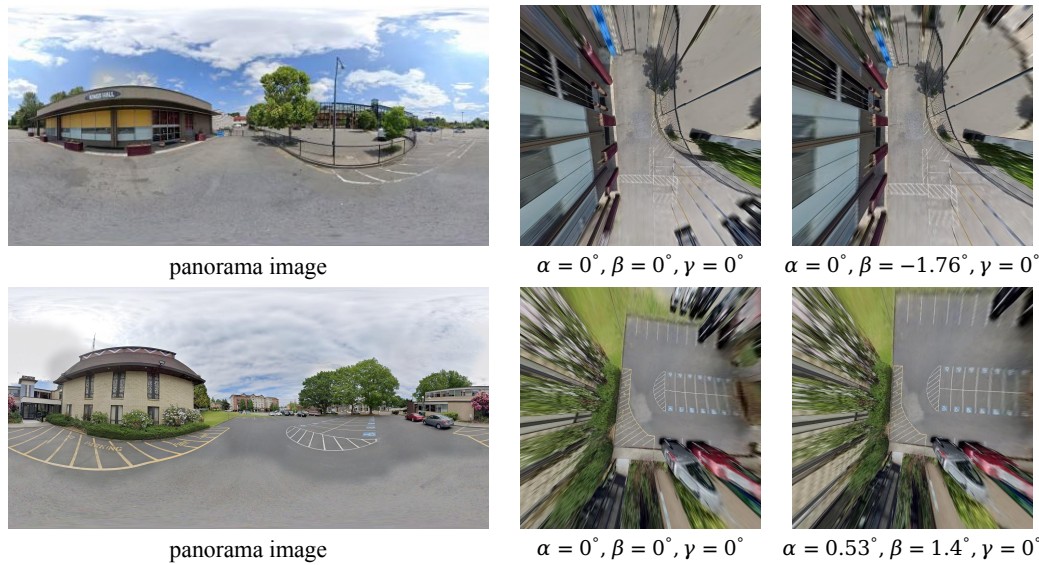

| panorama image | $\alpha = 0°, \beta = 0°, \gamma = 0°$ | $\alpha = 0°, \beta = -1.76°, \gamma = 0°$ |
| panorama image | $\alpha = 0°, \beta = 0°, \gamma = 0°$ | $\alpha = 0.53°, \beta = 1.4°, \gamma = 0°$ |

Figure 12: Bird's-eye view images obtained under different roll, pitch, and yaw angles using our spherical transform method.

## D   Broader Impact

This paper introduces an innovative approach to achieving high-precision localization of ground cameras, a capability of immense value in applications such as autonomous driving, augmented reality, and mapping services. Accurate localization is crucial for these industries to operate safely and effectively. However, it is imperative to acknowledge that the position of a vehicle or a user's camera is often sensitive and private information. The misuse or abuse of this technology could lead to privacy violations, potentially endangering individuals' security and personal information. We advocate safeguards and regulations to ensure that the privacy and security of individuals are not compromised in the process.

