# OpenReview forum: "Fine-Grained Cross-View Geo-Localization Using a Correlation-Aware Homography Estimator"
_NeurIPS.cc/2023/Conference — NeurIPS 2023 poster_

### Official Review · Reviewer_Ak83 · 2023-07-01

**Soundness:** 4 excellent
**Presentation:** 3 good
**Contribution:** 3 good
**Rating:** 6
**Confidence:** 5

**Summary:**

This paper propose to wrap the ground-view image to align with the corresponding aerial-view image using homography estimation. Firstly, a differentiable spherical transform is adopted to align the perspective between the ground-view and aerial-view images. Then a correlation-based homography estimator is proposed to align the similar parts of the transformed ground image and aerial-view image. It achieves state-of-the-art performance on VIGOR and KITTI datasets with a speed of 28 FPS.

**Strengths:**

+ The idea is very interesting and the writing is good.
+ The presentation of figures and tables are also good.
+ The evaluation is comprehensive and the performance is promising.


**Weaknesses:**

- The figure looks good, but I am not able to get the detailed flow of data. The part 2 and part 3 of Fig 2 are connected only on the correlation map. Would the code be released? It would be better to add this claim to help reproduce the result.
- The authors could add some insight about why the spherical transform is better than polar transform/ projective transform in the other papers.


**Questions:**

See limitations and weaknesses. I am open to raise my rating after rebuttal.

**Limitations:**

- The spherical transform is applied on query imaged on-the-fly, which might be a disadvantage compared to other transforms on aerial-view images, but the method is still fast. Is there a speed comparison with other methods? I think this could a limitation that needs to be discussed.
- The authors could add more discussion on positive/negative societal impact in supplementary materials. I think this method should have some potential impact on industry.

---

> ### Author Rebuttal · Authors · 2023-08-07
>
> **Q1: The detailed flow of data between part 2 and part 3 of Fig 2 and code.**
>
> **A1:** Thank you for the thoughtful question. The mentioned "part 2" refers to the computation of the homography matrix between the BEV image and its corresponding satellite image, denoted as $H^k$ in the diagram. The resulting matrix is subsequently passed to "part 3" for the computation of the loss.
> In "part 3," the final homography matrix obtained from "part 2" is denoted as $H^{itr}$.
>
> We appreciate your interest. We will release our codes.
>
> **Q2: Why the spherical transform is better than the polar transform/ projective transform.**
>
> **A2**: Thank you for your constructive suggestion. Firstly, our spherical transform projects the ground image to a bird's eye view, while polar transform or projective transform projects the satellite image to the perspective of the ground image (such as aligning with a panoramic view). There is a growing recognition and research on the role of Bird's Eye View (BEV) representation in localization, navigation, and related tasks.
>
> The advantage of projecting to a bird's eye view, as compared to projecting the satellite image to the perspective of the ground image, lies in its more intuitive nature for the localization task and the need for only one projection at the start of the localization process. On the other hand, the latter approach requires selecting projection points (initial pose or candidate poses) for projecting the satellite image, and if these points are not accurately chosen, the resulting projection may not align well with the ground image. Additionally, for more accurate localization, the latter approach requires multiple applications of the projection method from different positions, which could potentially make it less real-time compared to our method.
>
> Furthermore, in contrast to the polar transform methods, our spherical transform is derived from the imaging model of the ground camera, adhering closely to geometric principles, while the polar transform is considered a rough geometric alignment.
>
> In comparison to traditional Inverse Perspective Mapping (IPM) methods, our approach excels by not requiring any camera intrinsic or extrinsic parameters, thereby offering strong generality and versatility. It is precisely this advantage that allows us to project the panoramic images from the VIGOR dataset, which lacks camera calibration parameters, to a bird's-eye view perspective.
>
> **Limit.1: The speed comparison between spherical transform and other methods.**
>
> **A3**: We have evaluated the runtime of our spherical transform module.
> Our spherical transform module operates efficiently. Whether applied to the panoramic images in VIGOR or the frontal images in KITTI, the pixel correspondences between ground-level images and the bird's-eye-view can be precomputed during initialization. Subsequently, by employing `torch.nn.functional.grid_sample`, obtaining the corresponding BEV image takes under 1 ms for each processing instance.
> Furthermore, pixel projection calculations are matrix-based and GPU-accelerated using PyTorch, requiring less than 10ms to complete.
>
> After the acceptance of our paper, we plan to release our code as open source. Additionally, we will provide a Colab link for those who are interested in exploring and evaluating our spherical transform module.
>
> **Limit.2: Potential impact on industry.**
>
> **A4**: Thank you for your constructive suggestion. We will add potential impact on the industry in supplementary materials. Here we briefly discuss the positive impact of our method.
>
> We believe that our approach holds significant promise for a range of sectors, particularly in domains closely tied to autonomous driving, navigation, and geospatial technology. By introducing a novel pipeline, our method has the potential to redefine workflows within these industries, opening up exciting avenues for exploration and innovation.

---

> > ### Comment · Reviewer_Ak83 · 2023-08-15
> >
> > The rebuttal has addressed my concerns and I will keep my rating. I have checked the other reviews, but I do not find any ground for rejection.

---

> > > ### Author Response · Authors · 2023-08-15
> > >
> > > Dear Reviewer Ak83,
> > >
> > > Thank you very much for your insightful review and your decision to maintain the rating of acceptance! We are pleased to hear that our rebuttal has addressed your concerns.
> > >
> > > Should there be any further questions, please don't hesitate to let us know.
> > >
> > > Best regards,
> > > The Authors.

---

### Official Review · Reviewer_NjYP · 2023-07-04

**Soundness:** 2 fair
**Presentation:** 1 poor
**Contribution:** 2 fair
**Rating:** 3
**Confidence:** 5

**Summary:**

This paper addresses the problem of ground camera pose refinement by ground-to-satellite image matching. For this purpose, this paper proposes to project a ground image to the overhead view image plane by using a Homography, and then iteratively update the residual Homography between the projected overhead view image and the reference satellite image, from which the relative pose between the ground camera and the satellite image can be estimated. Experiments are conducted on two cross-view datasets, VIGOR and KITTI, and the results demonstrate the effectiveness of the proposed method.

**Strengths:**

+ A new pipeline for ground-to-satellite image localization using Homography update is proposed;
+ The proposed method achieves state-of-the-art performance.

**Weaknesses:**

(1) This paper estimates the residual Homography transformation between the reference satellite image and an overhead view image projected from a ground view image. The ground camera's pose is estimated from the Homography. However, no illustrations are about how to estimate the camera pose (rotation and translation) from the Homography.

One sentence in L205-206 is for orientation estimation. However, it seems that we should select different points from the ground view image and project them to the overhead view by using the estimated Homography, and the orientation is estimated by connecting the two points. If this is the way to estimate orientation, my first question is: "how to estimate the translation"? This is not described. My second question is, how to select the two points, randomly? Will this be reliable? The severe occlusion between the ground and satellite images makes that not all ground image pixels are visible in the overhead view and vice versa. How to select/determine points that are co-visible by the two views for this orientation estimation?

(2) If the pose is derived from the estimated Homography, it should be a deterministic solution. How the probability map in Fig.3 is estimated?

(3) The previous work by Shi and Li [22] proposes an iterative optimization network that directly optimizes pose parameters. From a high-level idea, the difference between this paper and Shi and Li is that this paper optimizes Homography parameters. However, the Homography contains 5 degrees of freedom which are larger than that of the pose parameters (3 DoF). What is the superiority of this? What is the performance of the proposed method if we simply modify the Homography parameter output to pose parameter output? and similarly, since this paper uses optical flow for supervision, why not directly estimate the optical flow between the two images and then compute the relative pose from the flow map?

(4) This paper claims that the GPS labels of the VIGOR and KITTI datasets are inaccurate and thus proposes to re-calibrate the pose labels. However, if the GPS labels themselves are inaccurate, the different GPS-to-UTM conversion methods would not correct the error.
Moreover, any investigations of what kind of GPS conversion methods are used in previous works when introducing the cross-view dataset, don't they also use the Mercator projection or an equivalent projection, or why the Mercator projection used in this paper should provide more accurate UTM coordinates than the methods used in their original papers?

Furthermore, if the ground truth has been modified, it is unfair to compare previous works' results reported in their original papers with the inaccurate poses for training and evaluation.

(5) The network is supervised using ground truth (GT) pixel correspondences. How are the GT correspondences derived, according to the GT relative RT? However, as I mentioned before, due to the severe occlusions between ground and satellite images, not every pixel in the ground view is visible in the overhead view and vice versa. Thus, there must be incorrectly derived GT correspondences if all the pixels in the overhead/satellite images are used.

(6) For the ground-to-satellite projection, this paper assumes the satellite image has a pin-hole camera projection with an FoV of 85-degree. However, Shi et al. [26] approximate a parallel projection for the satellite images. What is the superiority of this pin-hole camera projection over the parallel projection, and why use an FoV of 85 degrees instead of other numbers?

(7) The writing needs to be thoroughly improved and should be more accurate. For example:
    (i) Plz do not simply use a number [*] for references as a component in a sentence without mentioning the author's names.
    (ii) Sentence in L33 is not strictly correct. Slicematch is also a kind of "repeat (pose) sampling".
    (iii) Sentence in L144 "without any intrinsic parameters" is very ambiguous, this paper does use focal length and FoV for satellite images (L136-137), and the ground-to-satellite tomography-based projection should also use the ground camera's intrinsic when it is a pin-hole camera as in the KITTI dataset.
    (iv) The symbols in Sec 3.2 are very confusing. The authors mixed symbols for three images: the original ground image, the projected overhead view image from the ground-view image, the satellite image. This section needs to be thoroughly revised.
    (v) ...

(8) Missing important references:
     (i) Fervers, Florian, et al. "Uncertainty-aware Vision-based Metric Cross-view Geolocalization." Proceedings of the IEEE/CVF Conference on Computer Vision and Pattern Recognition. 2023. (available on arxiv in 2022, before SliceMatch [11])
     (ii) Shi, Yujiao, et al. "Where am i looking at? joint location and orientation estimation by cross-view matching." Proceedings of the IEEE/CVF Conference on Computer Vision and Pattern Recognition. 2020.
     (iii) Shi, Yujiao, et al. "CVLNet: Cross-view Semantic Correspondence Learning for Video-Based Camera Localization." Asian Conference on Computer Vision. Cham: Springer Nature Switzerland, 2022.
     (iv) Vyas, Shruti, Chen Chen, and Mubarak Shah. "Gama: Cross-view video geo-localization." European Conference on Computer Vision. Cham: Springer Nature Switzerland, 2022.
     (v) Zhang, Xiaohan, Waqas Sultani, and Safwan Wshah. "Cross-View Image Sequence Geo-localization." Proceedings of the IEEE/CVF Winter Conference on Applications of Computer Vision. 2023.


**Questions:**

Plz refer to my comments on Weaknesses

**Limitations:**

Using homography estimation between the ground and satellite images ignores the correspondences of objects not in the tomography plane, potentially limiting the performance.

---

> ### Author Rebuttal · Authors · 2023-08-07
>
> Thank you for your detailed reviews. Before addressing each individual question, we want to briefly introduce the pipeline of our proposed method to better clarify the technical details of our method.
>
> 1. We use a spherical transform to project ground images onto a bird's-eye view (BEV). Subsequent operations are on the BEV image and the satellite image only. The projection effect is shown in Figure 1(a)(b).
>
> 2. The Homography Estimator yields a matrix. With this matrix, we align the BEV and satellite images(Figure 1(b)(c)), creating overlap (Figure 1(d)).
>
> 3. **(3.1) Translation:** After aligning the two images, we can project the center point of the BEV image onto the satellite image (L189), derive its pixel coordinates, and obtain GPS via the inverse of Equation 4. The location of the BEV center point corresponds to the camera position. **(3.2) Orientation:** We project both the center point of the BEV image and another point along its vertical centerline onto the satellite image. Connecting these projected points establishes the camera's orientation. This process is analogous to determining a direction on a map by connecting two points: your current location and a point directly ahead on your path.
>
> 4. Training labels: Our method, like others, avoids direct GPS training. Instead, relies on pixel coordinate labels. Prior methods used meter-to-pixel resolutions, posing city-specific challenges. In contrast, our Mercator-based approach is universally applicable and accurate across cities.
>
> **Q1.1: How we estimate the camera pose (T, R) through Homography.**
>
> **A1.1:** See 3.1, 3.2 of the above pipeline.
>
> **Q1.2: "how to select the two points ... reliable? ... How ... co-visible by the two views ..."**
>
> **A1.2:** See 3.2 above for selecting points. Correlation-aware mechanism aligns correlated portions of BEV and satellite images, reducing non-visible area impact(L53-57).
> Co-visibility between points isn't strictly required for translation/orientation estimation via homography. So, there should not be a concern about the reliability of selected points. In Fig 1(d), the vehicle in BEV isn't visible in the corresponding satellite image. Using homography from visible parts alignment, we can align the vehicle's position on the satellite image and get its GPS.
>
> **Q2: How the probability map in Fig.3 is estimated?**
>
> **A2**: The Homography-derived pose is deterministically derived.
> The probability map doesn't affect pose estimation. Map gauges localization confidence, crucial for navigation tasks. It's a correlation map between the BEV center and satellite points, an intermediate in our network. Part 2 of Figure 2 illustrates map generation.
>
> **Q3: Why deriving Homography instead of direct pose? "Why not directly estimate the optical flow."**
>
> **A3**: Our network does not use optical flow supervision. We use a pair of pixel coordinates obtained from the localization point.
>
> Homography aligns images with geometry, uniquely solving pose. There is no need to learn Homography-to-pose transition. Our method can simplify networks and yield analytical solutions.
>
> Shi and Li's [22] needs camera parameters (Equation 3) for projection and invokes the Geometry Projection model iteratively. Our method doesn't need parameters and repetitive projection.
>
> Our data lacks optical flow supervision and optical flow estimation is inconsequential as we only use two points for translation and orientation.
>
> **Q4: "Claims that the GPS labels ... inaccurate" and "... provide more accurate UTM coordinate ..."**
>
> **A4:** We **do not** consider GPS labels problematic. Direct use of GPS info in training is challenging, necessitating pixel labels. VIGOR used uniform meter-to-pixel conversion across cities to calculate pixel-level labels from GPS, leading to significant errors(Sup. L73-75).
>
> SliceMatch highlights this, improving by calculating city-specific resolutions but still lacking precision. They mention in Sup.: "VIGOR dataset have used a ground resolution equal to 0.114m/pixel for all 4 cities" "We have calculated a new ground resolution for each city by averaging the ground resolutions."
>
> VIGOR and KITTI satellite images are from Google Maps, which uses Mercator. So Mercator is more accurate.
>
> SliceMatch fixed VIGOR labels, CCVPE uses them. Table 5 in our Sup. shows minor differences, averaging <0.2m vs. ours. Ours is universal and convenient.
>
> **Q5: "The network ... pixel correspondences. ... are used."**
>
> **A5**: For network supervision, we use one-pixel correspondence, not all image pixels. BEV center corresponds to camera GPS; Mercator gives pixel label from GPS.
>
> **Q6: "Assumes the satellite image has a pin-hole camera projection with an FoV of 85-degree."**
>
> **A6:** Our paper **do not** assume satellite imaging. We use the ground camera's imaging model for generating BEV. Our projection doesn't need repeated sampling for pose estimation like Shi et al.'s [26].
>
> The "FoV of 85 degrees" parameter affects BEV image FoV, as shown in Fig 7. We chose 85 degrees to achieve the optimal alignment with satellite map FoV (Sup. L34-37).
>
> **Q7: Writing.**
>
> **A7:** Thank you for your suggestion, but there are some misunderstandings as listed below.
>
> (i) Thank you. We will revise our paper.
>
> (ii) Our 'repeat sampling' means involving full process per pose, SliceMatch includes all candidate poses initially. So it is not "repeat sampling."
>
> (iii) No calibrated parameters are needed. See A6 and Sup. "A Projection Details."
>
> (iv) Sec 3.2 only alligns BEV and satellite images(subscripts 'g' and 's'). Mixed symbols are misunderstood. We didn't use the original ground image in sec 3.2.
>
> **Q8: References.**
>
> **A8:** We will add these papers in the revised paper.
>
> **Limit.1: "Using homography  ..."**
>
> **A9:** Our method handles discernible contours, even those not co-planar. Co-visible regions yield good outcomes. Experimental results show lower errors, confirming robustness and efficacy.

---

> > ### Comment · Reviewer_NjYP · 2023-08-16
> >
> > Thanks to the authors for the rebuttal.
> >
> > The response clears some of my concerns, while some still exist.
> >
> > > (3.2) Orientation: We project both the center point of the BEV image and another point along its vertical centerline onto the satellite image. Connecting these projected points establishes the camera's orientation.
> >
> > While this is reasonable, isn't there a closed-form solution for estimating rotation and translation from the homography matrix (e.g., by SVD decomposition [\*])?
> >
> > [\*] Malis, Ezio, and Manuel Vargas Villanueva. "Deeper understanding of the homography decomposition for vision-based control." (2007).
> >
> > What is the difference?
> >
> > > Q3: Why deriving Homography instead of direct pose?
> >
> > I am not convinced why not use the GRU to update the inplane relative rotation and translation (maybe also scale) between the BEV image and the satellite image. The in-plane relative rotation & translation & scale is equivalent to the Homography but with less degrees-of-freedom (DoF). Thus, they should be easier to learn compared to Homography. This relative rotation and translation is also the relative rotation and translation between the ground and the satellite image.
> >
> > > Q4: "Claims that the GPS labels ... inaccurate" and "... provide more accurate UTM coordinate ..."
> > I can understand the label correction for the VIGOR dataset. It is based on the assumption that the per-pixel distance (ground resolution) of satellite images for different region should not be constant. However, from my understanding, as long as the area is within the same UTM zone, the ground resolution is roughly the same. Thus, it is not clear how this correction will affect the performance. Furthermore, as in the initial comments, if the pose label in the original dataset is inaccurate, it is unfair to compare previous works' results reported in their original papers with the incorrect poses for training and evaluation. At least the authors should select the most state-of-the-art and re-train & evaluate it with corrected labels by this paper.
> >
> > There is also a label creation illustration for the KITTI dataset in the supplementary material. Although the steps sound reasonable, my confusion is: doesn't the original cross-view KITTI dataset provide GPS labels for ground and satellite images? Otherwise, how did previous works train and evaluation on this dataset?
> >
> > > Q6: "Assumes the satellite image has a pin-hole camera projection with an FoV of 85-degree."
> >
> > >A6: Our paper do not assume satellite imaging. We use the ground camera's imaging model for generating BEV. Our projection doesn't need repeated sampling for pose estimation like Shi et al.'s [26].
> >
> > >The "FoV of 85 degrees" parameter affects BEV image FoV, as shown in Fig 7. We chose 85 degrees to achieve the optimal alignment with satellite map FoV (Sup. L34-37).
> >
> > The BEV image assumes pin-hole camera projection with an 85-degree FoV. However, if the scene is not a planar and the BEV image projection differs from the satellite image projection, the relative transformation between the BEV image and the satellite image cannot be assumed as a homography.
> >
> > The satellite image is often approximated as an orthogonal projection. Thus, a straightforward understanding would be that the BEV image should also be approximated as an orthogonal projection. What is the problem with this? What is the superiority of the pinhole camera projection assumption over this orthogonal projection?
> >
> > For now, I keep my original rating.

---

> > > ### Author Response · Authors · 2023-08-17
> > >
> > > Thank you for your further inquiries. We are committed to addressing all the concerns you have raised.
> > >
> > > **Q1: Our method Vs SVD decomposition**
> > >
> > > **A1:**
> > > - SVD decomposition, as used to obtain camera pose (R, t), requires intrinsic camera parameters, per Eq (1) and (2) in Section 2.2 of the reference. Yet, these parameters are not available for our BEV and satellite images.
> > > - SVD decomposition of the homography matrix provides pose (R, t) between cameras, yet the lack of depth information results in scale ambiguity, limiting accurate ground camera position estimation.
> > > - Our approach is intuitive and efficient, offering a superior alternative to the complex and potentially time-consuming SVD decomposition.
> > >
> > > **Q3: Less DoF Vs Homography**
> > >
> > > **A3:** The transformation between BEV and satellite images encompasses more than 3 DoF. In practice, there may be potential roll and pitch deviations in addition to translation and yaw. The homography matrix accommodates these additional degrees of freedom in perspective transformation, rendering it a more realistic and robust choice for our scenario.
> > >
> > > Our homography estimation aligns correlated portions of BEV and satellite images. Co-visibility between points isn't strictly required for translation/orientation estimation via homography. This characteristic enhances the robustness of our method. For instance, in Fig 1(d), the vehicle in BEV isn't visible in the corresponding satellite image. Using homography, we can **align the vehicle's position on the satellite image and get its GPS.** This capability also extends the method's applicability.
> > >
> > > **Q4: VIGOR Label Correction**
> > >
> > > **A4:** The per-pixel distance of satellite images varies by region, evident from **SliceMatch's Supp.'s Table 4** and our Supp.'s Table 5.
> > >
> > > The labels we computed are very close to the corrected labels from SliceMatch and CCVPE (average error < 0.16m). We conducted experiments using their corrected labels for training, and the results showed that our label correction **did not significantly affect model performance** (mean dist errors on val data after augmentation were 3.55428m and 3.55148m, respectively). However, in order to **ensure the rigor of research and facilitate future research**, we still revised the data.
> > >
> > > Our method's merit lies in its **generality and convenience**. Unlike SliceMatch and CCVPE, our approach does not require city-specific resolutions. The following is the code of CCVPE:
> > >
> > >     if city[batch_idx] == 'NewYork':
> > >         meter_distance = pixel_distance * 0.113248 / 512 * 640
> > >     elif city[batch_idx] == 'Seattle':
> > >         meter_distance = pixel_distance * 0.100817 / 512 * 640
> > >
> > > However, we appreciate your suggestion and have conducted experiments by retraining and evaluating the most SOTA model (CCVPE) using our corrected labels. The results demonstrate that our labels do not noticeably impact model performance.
> > >
> > > **Q5: Train label creation for the KITTI**
> > >
> > > **A5:** The original cross-view KITTI dataset provides GPS labels for both ground and satellite images. In the absence of noise, the center pixel coordinates of satellite images serve as the ground truth label for the position of ground camera.
> > >
> > > Previous methods randomly displaced and rotated satellite images to generate pose labels. We followed a similar procedure during training and evaluation. Additionally, our model requires a pair of matched ground and satellite image pixel points. The GPS of the ground image in KITTI refers to the camera's optical center, which is not visible in its front-view image. Therefore, based on the camera's calibration parameters and point cloud data, we calculated the GPS coordinates corresponding to a pixel point in the BEV image. Then, using the Mercator method, we calculated the pixel coordinates of the GPS on the satellite map.
> > >
> > > **Q6: Pinhole camera projection for ground camera & orthogonal projection for satellite image**
> > >
> > > **A6:** Our goal is to estimate the corresponding pixel points of each pixel on the BEV imaging plane based on the ground camera's imaging model. Thus, we need to choose a method that is **more in line with the ground camera imaging model**, namely the pinhole imaging model.
> > >
> > > **In Shi et al., "Accurate 3-DoF Camera Geo-Localization via Ground-to-Satellite Image Matching" [26], Fig 5** illustrates both ground camera and satellite image imaging models. The illustration reveals that the ground camera's BEV obtained through the pinhole imaging model and the satellite image derived via orthogonal projection genuinely share the same viewpoint. Both satellite and our BEV images **project real-world points to a BEV imaging plane**.
> > >
> > > Building upon this imaging approach, imagine a concave semicircular surface on the ground; even though it is not a planar, BEV and satellite images can still be aligned through homography. Furthermore, our method has achieved SOTA mean localization accuracy, which demonstrates its robustness.
> > >
> > > Please inform me if you have any additional concerns.

---

> > > ### Author Response · Authors · 2023-08-20
> > >
> > > Dear Reviewer NjYP,
> > >
> > > We greatly appreciate your previous inquiries. As the discussion phase is nearing its conclusion, we would like to confirm whether we have adequately addressed your questions. Should you have any further inquiries or require additional clarification, please do not hesitate to inform us.
> > >
> > > Best regards,
> > >
> > > The Authors

---

> > > > ### Comment · Reviewer_NjYP · 2023-08-21
> > > >
> > > > Dear authors,
> > > >
> > > > I appreciate the further response.
> > > >
> > > > For Q3 and Q6, I am not fully convinced without experiment verification.
> > > >
> > > > For Q4, if the performance difference is insignificant, this aligns with our general intuition. Then, it is hard to confirm that the label correction method claimed in this paper is one of the novel contributions.
> > > >
> > > > For Q5, the GPS provided by the data refers to the GPS device, not the ground camera. The dataset provides the relative transformation between the GPS device and the left-ground camera. Thus, the ground camera's geo-location can be directly computed. This is what previous works do. Therefore, I don't find it necessary to re-create the camera pose labels from the point cloud is neccessary, if no evidence to demonstrate that the method for computing the ground camera's pose used in previous works is wrong.

---

> > > > > ### Author Response · Authors · 2023-08-21
> > > > >
> > > > > Dear Reviewer NjYP,
> > > > >
> > > > > Thank you for your further inquiries. We will strive to address them.
> > > > >
> > > > > **Q3: Less DoF vs Ours**
> > > > >
> > > > > **A3:** **Our paper's core goal is to align BEV perspective of ground and satellite images.** This alignment leverages perspective transformation through a homography matrix, rooted in multi-view geometry.
> > > > >
> > > > > We also appreciate your suggestion to compare our method with direct pose estimation approaches. We found a recent publication (first posted on arXiv on July 16, 2023) **that directly outputs pose (R, t)** and utilizes the same VIGOR and KITTI datasets as our paper:
> > > > >
> > > > > **[Ref 8] Yujiao Shi, et al. Boosting 3-DoF Ground-to-Satellite Camera Localization Accuracy via Geometry-Guided Cross-View Transformer.**
> > > > >
> > > > > According to the experimental data provided in [Ref 8], our method **exhibits significant advantages compared to theirs.** The details can be seen in the following table:
> > > > >
> > > > > Table 1 (Comparison results on VIGOR with aligned orientation.):
> > > > >
> > > > > Method| Area| ↓Mean(m)| ↓Median(m)|
> > > > > :--| :--| :--| :--|
> > > > > [Ref8]| Same| 4.12| 1.34|
> > > > > Ours| Same| **3.36(↓ 0.76)** | 1.36(↑ 0.02) |
> > > > > [Ref8]| Cross| 5.16| 1.40|
> > > > > Ours| Cross| **3.96(↓ 1.2)**| 1.68(↑ 0.28) |
> > > > >
> > > > > Table 2 (Performance comparison on KITTI with 20${}^{\circ}$ orientation noise):
> > > > >
> > > > > Test 1 corresponds to the same area, and Test 2 corresponds to the cross area.
> > > > >
> > > > > Method| Area   | ↑Lateral R@1m (%) | ↑Lateral R@5m (%)| ↑Long. R@1m (%) | ↑Long. R@5m (%) |
> > > > > :--| :--| :--| :--| :--| :--|
> > > > > [Ref8]| Test 1 | 76.44| 98.89 | 23.54| 62.18|
> > > > > Ours| Test 1 | **98.09(↑ 21.65)** | **100.0(↑ 1.11)** | **89.37(↑ 65.83)**| **99.31(↑ 37.13)**|
> > > > > [Ref8]| Test 2 | 57.72| 91.16 | 14.15| 45.00|
> > > > > Ours | Test 2 | **65.36(↑7.64)**| **96.02(↑4.86)**| **54.33(↑ 48.18)**| **80.32(↑ 35.32)**|
> > > > >
> > > > > **Q4: VIGOR Label Correction**
> > > > >
> > > > > **A4:** Even though experiments indicate that the impact of Label Correction on model performance is minimal, correcting the labels of the training set remains crucial. **Such a practice is in line with the spirit of scientific rigor.**
> > > > >
> > > > > In addition, the contribution of this Mercator-based method lies in **facilitating further related research (making it more general and rigorous) and the practical deployment of related models.**
> > > > >
> > > > > **Q5: Train label creation for the KITTI**
> > > > >
> > > > > **A5:** We are well aware that the GPS provided by the data refers to the GPS device and not the ground camera.
> > > > > The previous explanation was **provided for your understanding, as during training, the ultimate goal is to align the center pixels of satellite images with the ground camera's GPS.** Even in previous methods, the ground camera GPS was not directly computed but achieved through a displacement of the satellite image.
> > > > > The code snippets from HighlyAccurate and CCVPE are as follows:
> > > > >
> > > > >     sat_align_cam = sat_rot.transform(sat_rot.size, Image.AFFINE,
> > > > >                                             (1, 0, utils.CameraGPS_shift_left[0] / self.meter_per_pixel,
> > > > >                                              0, 1, utils.CameraGPS_shift_left[1] / self.meter_per_pixel),
> > > > >                                             resample=Image.BILINEAR)
> > > > >
> > > > > **We do not consider the previous methods problematic,** as we used the same GPS labels and the method of shifting the satellite map as they did. However, our proposed method does not use GPS labels during training. Instead, we use **a pair of corresponding pixels in the BEV and satellite images.**
> > > > >
> > > > > **Q6: Imaging Model & Non-planner Scene**
> > > > >
> > > > > **A6:**
> > > > > Regarding projection model choice, it is employed to establish **a correspondence between real-world spatial points and corresponding pixel coordinates in the image.** We adopted a pinhole imaging model that closely resembles real-world ground camera imaging, and there is no need for an experimental comparison between the pinhole imaging model and orthogonal projection. This is because, based on our principles, using orthogonal projection for BEV projection is not feasible.
> > > > >
> > > > > Should you be concerned about our method's robustness in non-planner scenarios, we'll provide proof in Supp to validate its capability to handle such situations.
> > > > >
> > > > > Our method **aligns correlated portions of BEV and satellite images**. As a result, our network can function effectively as long as alignable components are present in the scene.
> > > > >
> > > > > Because the vehicle is **driving on the road, the scene generally meets the planner's requirements,** we use a challenging scenario where the positioning point is obstructed by a higher-level bridge for verification.
> > > > >
> > > > > In the VIGOR scene 'Seattle/panorama/-8rg8KItuIfeaLAg6G_gxQ,47.571477,-122.337678,.jpg', even with obscured localization due to an overhead bridge (visible on Google Maps at <47.571477,-122.337678>) our method can still successfully achieve localization, with an error of only 1.28 meters.

---

### Official Review · Reviewer_ZVTo · 2023-07-06

**Soundness:** 3 good
**Presentation:** 2 fair
**Contribution:** 2 fair
**Rating:** 3
**Confidence:** 4

**Summary:**

The paper addresses fine-grained cross-view geo-localization task that that matches the camera ground images with a satellite image patch covering the same area to determine the geo-pose of camera. The proposed approach projects ground images onto a bird’s-eye view perspective and formulate the task as a 2D image alignment problem. A correlation-based homography estimation module is proposed to achieve precise localization. Experiments are presented on VIGOR and KITTI datasets.

**Strengths:**

+ Proposed correlation-aware homography estimation module is interesting
+ Experiments show promising results in VIGOR and KITTI datasets.

**Weaknesses:**

The paper has interesting ideas, but I think the paper needs more work to be convincing. The weaknesses are mentioned below:

-- The use of BEV representation has been explored for  Fine-Grained Cross-View Geo-Localization task [Ref1]. This paper should have been discussed and compared in the experiments.

[Ref1] F. Fervers, et. al., Uncertainty-aware Vision-based Metric Cross-view Geolocalization, CVPR 2023


-- The novelty behind the spherical transform module to project ground images to a birds-eye-view (BEV) perspective is not clear. Inverse Perspective Mapping (IPM) is commonly used for long time for transforming camera images to BEV [Ref2, Ref3]. Also, looking at the generated BEV images and from experience of using [Ref3] for generation, the BEV images do not seem any better than the quality achieved by applying the IPM technique.

[Ref2] H. A. Mallot, et. al., “Inverse perspective mapping simplifies optical flow computation and obstacle detection,” Biological Cybernetics, 1991

[Ref3] L. Reiher, et. al., "A Sim2Real Deep Learning Approach for the Transformation of Images from Multiple Vehicle-Mounted Cameras
to a Semantically Segmented Image in Bird’s Eye View", IEEE International Conference on Intelligent Transportation Systems, 2020

-- There are also many works that use neural networks to generate BEV representation from ground camera images. Some of these papers should be discussed. (e.g., [Ref4, Ref5])

[Ref4] Z. Li, et. al., Translating Images into Maps, ICRA 2022
[Ref5] A. Saha, et. al., BEVFormer: Learning Bird’s-Eye-View Representation from Multi-Camera Images via Spatiotemporal Transformers, ECCV 2022

-- The utilization of infoNCE loss for cross-view image geo-localization is also not new [Ref6, Ref7]

Y. Zhu, et. al., Simple, Effective and General: A New Backbone for Cross-view Image Geo-localization, arXiv 2023
F. Deuser, et. al., Sample4Geo: Hard Negative Sampling For Cross-View Geo-Localisation, arXiv 2023


-- Experiments do not show consistent improvements, as evident from Table 2. There are several cases where other baselines perform better.

-- Not sure of the reproducibility of this work and no code is provided in the supplementary.

**Questions:**

Please see the weaknesses.

I have read the author’s rebuttal and other reviews. I am still not convinced and keep the score the same.

---

> ### Author Rebuttal · Authors · 2023-08-07
>
> **Q1: "The use of BEV representation has been explored ..."**
>
> **A1:** Sorry for the confusion. We'd like to clarify a point. The **BEV (Bird's Eye View) images** used in our paper are explicitly derived by exploiting the geometry of the scene. In contrast, the references [Ref1] and others utilize networks to learn how to project other-view image inputs to BEV representations at the **feature level**. It's important to note that none of these references employ explicit BEV images resembling Fig. 1(b) presented in our paper.
>
> While some research focuses on learning BEV representations within the BEV space, we explore the use of BEV images for Fine-Grained Cross-View Geo-Localization. In the Related Work of [Ref 1], it is explicitly mentioned that "PV2BEV (the perspective view to bird's eye view transformation) methods can be categorized based on whether they explicitly exploit the geometry of the scene to bridge the gap between PV and BEV or learn the mapping in a data-centric manner." They follow the second approach, using a spatiotemporal transformer encoder for multi-camera/timestamp to BEV mapping.
>
> Our method uses the first approach. Our proposed pipeline specifically requires BEV images for direct alignment with Aerial imagery, rather than employing a high-level BEV representation (which exists in the feature space).
>
> As evident from Figure 1 we provided, our BEV method intuitively enhances information alignment with aerial imagery, reducing network complexity.
>
> We plan to expand the discussion on various BEV presentation methods in the related work section. However, direct comparisons may not be feasible due to fundamental differences in our BEV approach compared to [Ref 1]. Furthermore, [Ref 1] uses datasets(KITTI-360) with multiple vehicle cameras for joint localization involving temporal data. In contrast, our method directly localizes using single panoramic(VIGOR) or frontal-view images(KITTI) aligned with satellite maps.
>
> **Q2: The novelty and advantage of our spherical transform compared with IPM.**
>
> **A2**: Sorry for the confusion. Our spherical transform achieves high-quality BEV images from ground images comparable to IPM without the need for the camera's intrinsic and extrinsic parameters, detailed in the 'A Projection Details' section of Supplementary Materials.
>
> As mentioned in [Ref1]'s related work, "IPM transforms PV features to BEV via a homography based on the camera’s intrinsic and extrinsic parameters." The IPM method requires knowledge of the camera's intrinsic and extrinsic parameters, which are not available for the VIGOR dataset. Our proposed spherical transform module can project ground images to a BEV perspective without the need for any camera’s intrinsic and extrinsic parameters.
>
> In conventional IPM, distinct parameters are required for calculating BEV images in distinct cities, tailored to each camera setup. In contrast, our spherical transform enables a unified approach and interface for projecting BEV images.
>
> Moreover, our spherical transform operates efficiently. The pixel correspondences between ground-level images and BEV can be precomputed during initialization. Obtaining the BEV image takes under 1 ms for each processing instance. Moreover, pixel projections are matrix-based and GPU-accelerated in PyTorch, completing in <10ms.
>
> **Q3: "There are also many works that use neural networks to generate BEV representation ..."**
>
> **A3**: Please refer to Q1 for the distinction between BEV representation and the BEV image derived from our method. We will add discussions of the differences between BEV representation and our method in our revised paper.
>
> Also, there is a significant distinction between our explicit method for obtaining BEV images and the BEV representation approaches[Ref1, Ref4, Ref5]. The significance of our method lies in its ability to operate without the need for camera intrinsic and extrinsic parameters, coupled with its rapid processing speed. This allows for seamless integration into applications employing BEV representation methods.
>
> **Q4: "The utilization of infoNCE loss for cross-view image geo-localization is also not new."**
>
> **A4:** The utilization of infoNCE loss is not the main contribution of our work. Despite InfoNCE loss having been previously explored ([Ref6, Ref7]), we share different motivations.
>
> We highlight its role in homography estimation given limited point supervision. It maximizes label utilization during training. As stated in our introduction, homography estimation typically requires a minimum of four corresponding point pairs for accurate computation. However, the majority of datasets we work with only provide a single pair of points as supervision (i.e., the localization position).
>
> Furthermore, in Section 4.5, we have also indicated that our model training can be accomplished even without this loss, yet still yield competitively robust outcomes (Table 3).
>
> **Q5: "Experiments do not show consistent improvements, as evident from Table 2."**
>
> **A5**: The primary discrepancy in performance in the original paper lies in the orientation estimation. Experimental results demonstrate our model's capability to estimate orientation even without the requirement for orientation labels during training. The introduction of orientation supervision can lead to a significant improvement in the performance of orientation estimation.
>
> We include an updated Table 2 after incorporating orientation supervision. Please refer to the global response and Table 1 in the uploaded PDF for more details.
>
> **Q6: Reproducibility of our work.**
>
> **A6**: We will release our code, model checkpoints, and training scripts.

---

> > ### Comment · Reviewer_ZVTo · 2023-08-15
> > **Comment by Reviewer ZVTo**
> >
> > Thanks for the Rebuttal. However, I still hold my original opinion. I think detailed analysis and comparison with other geometry-based and learning-based BEV methods are critical, which is missing. About IPM compared to the proposed spherical transform approach, I still do not think there is a significant enough difference. IPM can utilize the actual camera's intrinsic and extrinsic parameters. However, IPM will also be able to utilize image height width and 85-degree field-of-view (as used in the paper) to estimate an intrinsic. The extrinsic used are relative and help IPM more when the BEV is generated using multiple images with displacement from the center.

---

> > > ### Author Response · Authors · 2023-08-17
> > >
> > > **Q1: 'I think detailed analysis and comparison with other ... BEV methods are critical.'**
> > >
> > > **A1:** Thank you for suggesting a comparison between our method and other geometry-based and learning-based BEV methods.
> > >
> > > While we appreciate the suggestion to conduct a comparative experiment with [Ref1], it's important to note that [Ref1] chose to create a new dataset for their study, **instead of utilizing the commonly used VIGOR or KITTI datasets** in the Fine-Grained Cross-View Geo-Localization field. This decision is likely due to two main factors:
> > >
> > > **1.** The need for multi-view ground cameras:
> > >   - "The ground image of the **i-th** camera is encoded into feature map F_{Gi}."
> > >
> > > **2.** The need to use true camera intrinsic parameters for memory and computational optimization. VIGOR's panoramic images lack this requirement:
> > >   - "The points are ... projected onto the camera plane **using its extrinsic and intrinsic parameters**."
> > >
> > > However, we find another recently accepted learning-based BEV method [Ref 8] from papers that cite [Ref1]. **This article is new**, posted on arXiv on July 16, 2023, and **uses the same VIGOR and KITTI datasets as us**:
> > > - [Ref 8] Yujiao Shi et al. "Boosting 3-DoF Ground-to-Satellite Camera Localization Accuracy via Geometry-Guided Cross-View Transformer"
> > >
> > > This paper explicitly describes the use of a transformer method to learn image representations in the BEV space.
> > >
> > > According to the experimental results provided in [Ref 8], our method **exhibits significant advantages in comparison.** Please refer to Table 1 and Table 2 below for specific comparison results:
> > >
> > > Table 1 (Comparison results on **VIGOR** with aligned orientation):
> > > Method  | Area   | ↓Mean(m)     | ↓Median(m)   |
> > > :------ | :----- | :----------- | :----------- |
> > > [Ref8]  | Same   | 4.12         | 1.34         |
> > > Ours    | Same   | **3.36(↓ 0.76)** | 1.36(↑ 0.02) |
> > > [Ref8] | Cross  | 5.16         | 1.40         |
> > > Ours    | Cross  | **3.96(↓ 1.2)**  | 1.68(↑ 0.28) |
> > >
> > > Table 2 (Performance comparison on **KITTI** with 20${}^{\circ}$ orientation noise, where Test 1 corresponds to the same area, and Test 2 corresponds to the cross area):
> > >
> > > Method  | Area   | ↑Lateral R@1m (%) | ↑Lateral R@5m (%)| ↑Long. R@1m (%) | ↑Long. R@5m (%) |
> > > :------ | :----- | :------------- | :------------ | :----------------- | :----------------- |
> > > [Ref8]  | Test 1 | 76.44          | 98.89         | 23.54              | 62.18              |
> > > Ours    | Test 1 | **98.09(↑ 21.65)** | **100.0(↑ 1.11)** | **89.37(↑ 65.83)**  | **99.31(↑ 37.13)** |
> > > [Ref8]  | Test 2 | 57.72          | 91.16         | 14.15              | 45.00              |
> > > Ours    | Test 2 | **65.36(↑7.64)**   | **96.02(↑4.86)**  | **54.33(↑ 48.18)**   | **80.32(↑ 35.32)**   |
> > >
> > > Furthermore, we would like to emphasize our method's motivation and the distinction between our approach and other BEV representation methods.
> > > - Our method aims to obtain 3-DoF poses of ground cameras by **aligning ground BEV images** with satellite images from the same viewpoint.
> > >   - So, what our method requires is an **explicitly obtained BEV image**
> > > - In contrast, methods such as [Ref1]:
> > >   - first employ an encoder to generate **feature maps** of ground images
> > >   - and then use transformers to construct the BEV representation based on these feature maps.
> > >     - using the ground image features as keys and values.
> > >   - The outcome of these methods is a feature map.
> > >
> > > **Q2: 'IPM will also be able to ... estimate an intrinsic. The extrinsic ... is generated using multiple images with displacement from the center.'**
> > >
> > >
> > > **A2:** Our spherical transform approach is motivated by the need to convert **panoramic images** into BEV images. To the best of our knowledge, no readily available IPM method fits our panoramic data. Thus, we derived and successfully implemented this method ourselves.
> > >
> > > Traditional IPM methods involve three coordinate systems: image's, camera's, and world's. The camera's unit of **measurement is pixels**, while camera's and world's use **meters**. Traditional IPM computes spatial coordinates in the world coords on the ground corresponding to image pixel coordinates, then obtaining BEV image. Our method directly computes **between the original image and the BEV image**, eliminating the need for coordinates in meters. This simplifies the process to a certain extent.
> > >
> > > Extrinsic parameters in IPM are not just for multiple images. The derivation of IPM for individual frames also requires the **camera's height above the ground as an extrinsic parameter** which is not available in VIGOR. For example, in MATLAB IPM routine(https://ww2.mathworks.cn/help/driving/ref/birdseyeview.html), they explicitly state the need for this parameter, stating: "Set the height of the camera to be about 2 meters above the ground." Similarly, in Bertozzi et al.'s 'Stereo Inverse Perspective Mapping: Theory and Applications,' Eq. (1), 'h' is the true camera height above the ground.
> > >
> > > Please let us know if you have further concerns.

---

> > > ### Author Response · Authors · 2023-08-20
> > >
> > > Dear Reviewer ZVTo,
> > >
> > > We greatly appreciate your previous inquiries. As the discussion phase is nearing its conclusion, we would like to confirm whether we have adequately addressed your questions. Should you have any further inquiries or require additional clarification, please do not hesitate to inform us.
> > >
> > > Best regards,
> > >
> > > The Authors

---

### Official Review · Reviewer_NnZj · 2023-07-09

**Soundness:** 3 good
**Presentation:** 3 good
**Contribution:** 3 good
**Rating:** 5
**Confidence:** 5

**Summary:**

The paper proposes a homography estimation-based method for cross-view geo-localization. Contrary to existing methods that tackle the problem as a retrieval problem, the paper proposes to reformulate the problem as homography estimation of aligning the birds-eye-view against the satellite image. To this end, the authors propose a recurrent homography estimation module which learns to estimate the homography matrix given the feature maps of birds-eye-view and satellite images. Experiments show that the method can demonstrate effective performance against the tested baselines in most metrics.

**Strengths:**

1. The idea of using bird's eye view for matching against satellite images is intuitive, and the proposed method well leverages the formulation by casting the geo-localization problem as homography estimation.

2. Experiments make valid comparisons against both existing geo-localization methods and also on a few image matching-based methods, where the proposed method shows performance enhancements. Notably, the paper shows large amounts of improvements in translation estimation, which I find to be highly practical for geo-localization tasks.

3. Writing is very clear. It was straightforward to follow the paper's motivation and how the proposed method functions as a whole.


**Weaknesses:**

My concerns are two-fold:
1. While I favor the paper's formulation of aligning bird's eye view against satellite images, I feel the paper has only included a limited number of homography-based baselines. Currently the paper has only made comparisons against homography estimation from local feature maches in Table 3. However there are many more homography estimation methods that could additionally be tested, for example 'Deep Image Homography Estimation' (which is also introduced in the related works section). Adding more homography-based methods for comparisons will better validate why the proposed recurrent homography estimator is needed. Related, how are the homographies estimated from SuperGlue, LoFTR matches? Have the authors used a RANSAC-style estimator to handle outliers? It seems that the paper is currently missing details on this part, which is crucial for local feature matching-based methods to work effectively.

2. I have not fully grasped the reason why the method performs poorly for rotation estimation in KITTI. Is this due to not using the orientation data labels during training? A better clarification of this phenomenon will be helpful for readers to better understant the paper.

**Questions:**

Please refer to the 'weaknesses' section.

**Limitations:**

The authors have not explicitly discussed limitations.

---

> ### Author Rebuttal · Authors · 2023-08-07
>
> **Q1.1: Comparison with other homography-based methods.**
>
> **A1.1:** Thank you for your constructive suggestion. In the introduction of our paper, we highlight that two significant challenges arise during homography estimation in the cross-view localization task. The first challenge pertains to the substantial presence of occlusions and ambiguous content in the scene (as indicated in L52-L53), while the second challenge is the lack of compact supervision information for homography estimation, i.e., a minimum of four matching point pairs (as stated in L59-L60).
>
> These challenges are why we did not employ other homography estimation methods. For instance, methods like "Deep Image Homography Estimation" require complete supervision. We also considered the use of unsupervised methods, where such techniques generally compute the similarity between two images to obtain a loss. However, our early experiments demonstrated that L1 loss (photometric similarity loss) and SSIM loss (Structural Similarity loss) were not very effective.
>
> Furthermore, our envisioned application scenarios, such as autonomous driving and robot navigation, demand high real-time performance from the network. Hence, methods like those mentioned in the "Related Work" section, which are based on GANs and transformers and involve substantial computational overhead, are not suitable.
>
> One of the most pivotal attributes of our proposed homography estimator lies in its correlation-aware mechanism. Our choice of the Correlation-Aware Homography Estimator is inherently well-suited to our task's demands, encompassing aspects such as weakly supervised training, adeptness in handling occluded scenes, and real-time capabilities.
>
> However, we will explore testing other advanced unsupervised Homography Estimation networks in the future.
>
> **Q1.2: How are the homographies estimated from SuperGlue, LoFTR matches?**
>
> **A1.2:** We employed the RANSAC method, specifically applying the findHomography function from the OpenCV library on the matches obtained from SuperGlue and LoFTR, utilizing the cv2.RANSAC parameter. We will add relevant information to the paper
>
> **Q2: Performance of rotation estimation in the KITTI dataset**
>
> **A2:** Thank you for the thoughtful question. Indeed, our method does not utilize orientation labels, unlike the other methods we compared against in Table 2 of our paper. L204-L206 in the main paper detail how we estimate orientations.
>
> Our experimental results highlight that our model can estimate orientation without orientation labels during training, particularly showing a marked improvement in estimation accuracy on the VIGOR dataset compared to previous methods. The diminished performance in orientation estimation on the KITTI dataset may be due to the presence of only front-view images, in contrast to VIGOR's panoramic images. This limited field of view in KITTI restricts the available information, leading to suboptimal performance. For instance, the projected BEV image covers less than one-fourth of the corresponding satellite image, as depicted in Fig 4(d), compared to over two-thirds in the VIGOR dataset, as shown in Fig 1(d).
>
> We have also tried using orientation labels to guide our model training, employing L1 loss of orientation error, which has led to substantial improvement in orientation estimation on the KITTI dataset. The table below illustrates the enhanced orientation performance compared to training without orientation labels. Even without meticulous hyper-parameter tuning, the orientation estimation has improved significantly. Specifically, under the "Same" and "Cross" settings, the mean orientation estimation error has decreased by 3.13 degrees and 3.92 degrees, respectively. We have also updated Table 2 to include orientation supervision; please refer to Table 1 in the PDF of the global response.
>
> |Area|Ori. Label| ↓Mean| ↓Median | ↑R@1$^\circ$  | ↑R@5$^\circ$ |
> | :--- | :---: | :---: | :---: | :---: | :---: |
> | Same| w/o. | 3.93 | 3.34 | 15.45 | 70.05 |
> | Same| w.  | 0.80(↓ 3.13) | 0.61(↓ 2.73) | 71.87(↑ 56.42) | 99.62(↑ 29.57) |
> | Cross| w/o. | 7.10 | 4.60 | 11.78 | 53.23 |
> | Cross| w.  | 3.18(↓ 3.92) | 1.94(↓ 2.66) | 27.83(↑ 16.05) | 82.21(↑ 28.98) |

---

> > ### Comment · Reviewer_NnZj · 2023-08-12
> >
> > Thanks for the additional set of experiments and clarifications. I am willing to keep my score.

---

> > > ### Author Response · Authors · 2023-08-15
> > >
> > > Dear Reviewer NnZj,
> > >
> > > Thank you very much for your recognition and your recommendation for acceptance. We appreciate your thoughtful review. Should you have any further questions or concerns, please don't hesitate to reach out. We are here to respond to your inquiries, as well as those of the other reviewers.
> > >
> > > Best regards,
> > > The Authors.

---

### Author Rebuttal · Authors · 2023-08-10

We extend our sincere thanks to all reviewers for their insightful feedback. Our method has been recognized as "interesting" (Reviewer Ak83, ZVTo), "new" (Reviewer NjYP), and "intuitive" (Reviewer NnZj). We are gratified that the "large amounts of improvements" (Reviewer NnZj), "state-of-the-art performance" (Reviewer NjYP), and "promising results" (Reviewer ZVTo, Ak83) of our method, along with the "comprehensive evaluation" (Reviewer Ak83), have been collectively acknowledged. The clarity of writing, as well as the presentation of figures and tables, have also been commended (Reviewer NnZj, Ak83).

We have addressed individual comments in the following sections and hope our responses satisfy any concerns.

---
**For Reviewer NnZj and Reviewer ZVTo**

In Table 1 of the newly uploaded PDF, we have included an updated version of Table 2 from our main paper, utilizing orientation labels (without meticulous hyper-parameter tuning). This update demonstrates significant improvement in orientation estimation. We will include relevant details in our revised paper.

We would like to draw your attention to the SliceMatch metrics, which were directly borrowed from the original paper. However, the two red-marked values are likely erroneous, as they are identical. The values for R@$1^{\circ}$ and R@$5^\circ$ should not be equal, and based on the comparison with our method on other metrics, it is likely that the value for R@$1^{\circ}$ is incorrect. We suspect this value may have been copied inaccurately.

Additionally, it's worth noting that CCVPE is our concurrent work and has not yet undergone peer review. Compared to SliceMatch (CVPR 2023), our model consistently exhibits substantial improvements across all metrics.

---
Thank you once again for all your constructive insights!

---

### Decision · Program_Chairs · 2023-09-21

**Decision:**

Accept (poster)

**Comment:**

Reviewers are split on the paper: weak accept, borderline accept, and 2 rejects. Authors provided extensive rebuttals but all the reviewers kept the original ratings even after the rebuttals and the discussions. The area chair looked at the paper and the reviews carefully. In the end, the area chair recommends acceptance for the following reasons: 1) The negative reviews are sometimes due to misunderstanding of the paper; 2) Many concerns of the reviewers appear nit-picking, where authors' rebuttal sounded more convincing; and 3) The other two positive reviewers are excited with the paper. Please read all the reviews carefully and prepare the camera ready by addressing them.